# Influence of agility training on body-size and object solidity perception in pet dogs

**Dario Starić** [1][*], **Lea Arnauer** [2], **Sarah Marshall-Pescini** [1], **Friederike Range** [1]

**1** Domestication Lab, Konrad Lorenz Institute of Ethology, University of Veterinary Medicine Vienna, Austria, **2** University of Vienna, Vienna, Austria

◉ Equal contribution and first authorship: these authors contributed equally to this work and share first authorship
* Dario.Staric@vetmeduni.ac.at

## Abstract

Several studies suggest that dogs are a suitable model to study body-awareness in a systematic and ecologically relevant manner. However, previous studies used a single door paradigm, which does not allow for the control of motivation and thus may yield false positive results. Moreover, it is unclear how life experience might influence body-awareness. In this study, we tested self body-size and -shape as well as object solidity perception in pet dogs and agility trained dogs in a two-choice paradigm. The subjects had to walk through one of the two openings in a fence to reach a reward. The openings were of different shape and/or size (Experiment 1) or covered by different materials (Experiment 2). Dogs could correctly assess their size, while the results are less conclusive for shape perception. They also showed some understanding of the solidity of materials. Agility dogs took less time to reach the reward in Experiment 1, but took more time in Experiment 2. They also showed better shape perception. These results show that life experience has an effect on body-awareness.

## Introduction

When navigating their surroundings, animals require a basic understanding of the physical properties of objects around them and, to some extent, at least the most basic level of self-awareness (i.e., distinguishing between external objects and their own self) to avoid other objects, hazards, or obstacles [1]. In addition, they also need the notion of their own body proportions (i.e., size, shape, weight, etc.) to master this challenge. For instance, an animal has to have some knowledge of its size to avoid being stuck in a narrow opening or trying to jump over a gap that is too large, but also an understanding of what materials can support their weight to avoid falling. This level of self-awareness is usually called the "ecological self" and is one of the five "selves" or categories used to describe self-awareness [2]. Following the idea self-awareness is achieved through several cognitive processes [1,3], we can further break down the ecological self into smaller "building blocks".

**Data availability statement:** All relevant data are within the manuscript and its Supporting Information files.

**Funding:** This research was funded in whole or in part by the Austrian Science Fund (FWF) grants DOI 10.55776/P33928 and 10.55776/W1262 (received by FR). For open access purposes, the author has applied a CC BY public copyright license to any author accepted manuscript version arising from this submission. Funders had no role in study design, data collection and analysis, decision to publish, or preparation of the manuscript.

**Competing interests:** The authors have declared that no competing interests exist.

The first "building block" of perceiving oneself as an "object" or an "obstacle" in the physical world entails that, when moving an object in their environment, an animal needs to be aware of its own position in relation to the object, so as not to obstruct the trajectory of the object. A more social example is that animals living in groups with a dominance hierarchy might need to consider their position in relation to other members of the group, as crossing paths with a more dominant member of their group or blocking a way to a food source might cause conflicts.

The next level or building block involves not only the understanding that the body acts as an object, but also that it has certain physical properties (e.g., size, shape, weight...). Perhaps the most important property of the self as an object is its size. Misjudging one's own size can lead to perilous situations for animals on all trophic levels, either by getting stuck in their environment, attacking a too large prey or rival, or failing to find an appropriate hiding place. Finally, relating one's own physical properties to those of the objects in their environment forms the third building block. For example, arboreal animals should judge the elasticity of tree branches in relation to one's own weight to reduce the risk of falling. Thus, only understanding the first two building blocks of the ecological self is not sufficient to make adequate predictions about the consequences of one's actions. However, the building blocks may vary between species as different species experience them through different senses and different blocks may have different ecological relevance for each species [3,4]. Furthermore, individual experience affects the understanding of each building block [5]. For instance, an animal living in a more crowded environment (e.g., a bird species living in thick forests) will have more information about its body through constant interaction with the objects around them, while an animal living in a more open space may have limited feedback on its body (e.g., a bird species living on planes). This may lead to a false perception of self and wrongly estimated physical attributes of self may affect multiple building blocks (e.g., crayfish raised in smaller shelters over-estimate their size, which in turn makes them over-estimate their fighting potential [6]).

A paradigm used to test for the understanding of **ones' body as an obstacle** consists of a mat or a carpet that the subject is standing on and a moveable object/toy that is attached to it. The subject is then requested to pass/give the object to the experimenter. If the subject realizes it has to move off the mat in order to give the object (together with the mat) to the experimenter, we assume that they have the ability to understand that their body is a physical object and can act as an obstacle. This paradigm was originally created to test toddlers [7], but to our knowledge only two studies used it in non-human animals: dogs and elephants [8,9]. Both elephants and dogs have shown an understanding of their own body as an obstacle by moving from the mat significantly more when the objects were attached to it than in the control condition(s).

To our knowledge, only a few studies investigated the ability to put the **properties of one's body in relation with the properties of other objects** in their environment [1,10–17]. Awareness of one's **own body size** is the most commonly studied **property of body as an object**, and it is usually studied in the context of species whose males fight for resources (territory, potential mates, etc.) (e.g., [6,18]). In that context,

it is beneficial for an individual to have a perception of one's own relative size to be able to assess how much bigger or smaller the opponent is. Only few studies so far tested body size perception in the context of foraging and moving through the environment [1,10–17]. The paradigm most frequently used is presenting the animals with gaps of different sizes to go through or walk over. If the animals are aware of their body size, they should choose to walk through bigger gaps when available, adjust their bodies accordingly as the gaps get smaller, and be more hesitant to attempt going through small gaps. Alternatively, some studies used a single gap paradigm in which case the latencies to approach or reach the reward and the behavioural modifications (e.g., body pose, ducking the head, etc.) were taken as the measure of the awareness of body size. Animals across taxa seem to be aware of their body size and adjust their behaviour as they move through or walk over differently sized gaps [1,5,10–12], suggesting that understanding of one's own size is indeed crucial for successful navigation through the environment. The only study that focused on other properties of body as an object (other than shape and size) was carried out on brown rats [13]. The rats had to cross one of the three bridges presented in the enclosure to reach a reward, however only one bridge was fixated, while the other two were loose (acting like a see-saw). The brown rats seem to be able to put their own body weight in the relation to the solidity of the object they walk on. However, out of 25 animals, only 8 developed a probing behaviour (tapping the bridge with their paws) to discriminate between bridges, 12 gave up after a few falls, and the remaining 5 did not develop the correct strategy but kept on trying. This raises an interesting point that although the rats may be aware of their weight and how it interacts with a loose bridge, it does not mean that animals develop the correct strategy to test the properties of another object in the environment and, as mentioned by the authors, some rats might have perceived the falling off the bridge as too big of a risk to invest more energy to finding a solution.

Although a number of studies have been carried out on different species to test aspects of body self-awareness, no single species has been systematically tested in all "building blocks", making it difficult to progress from an approach where individual abilities are being tested, to one in which we are also able to further our understanding of how these building blocks integrate and complement each other. Lenkei and colleagues [1] proposed that dogs would be an optimal model to systematically investigate self-awareness across all building blocks, define the different "building blocks", and further our understanding of the underlying cognitive processes required to achieve them. The first two building blocks mentioned above have already been tested in dogs. Dogs have been shown to understand that their body acts as an obstacle [9] and that they can correctly judge their body size [1,10]. The study by Lenkei and colleagues [1] also focused on shape perception by comparing dog breeds with different leg lengths (short legged vs long legged breeds) and found that leg length did not affect the assessment of suitability of an opening to walk through. However, although leg length contributes to the overall shape of the body, the legs may not be the main limiting factor when assessing whether the body can fit through an opening. Finally, to our knowledge, the third building block mentioned (i.e., putting properties of one's body in relation with the properties of other objects) above has never been tested in dogs.

Dogs occupy a special environmental niche – the obstacle-rich, complex human-environment– and partake in an array of human-dog interactions with some of them providing extensive opportunities to learn about their body size and shape (e.g., agility, search & rescue, musical canine freestyle dogs). Proprioception and exceptional body-awareness is especially important in agility dogs as they have to decide within milliseconds how to move their body to not throw off a bar at a jump, balance on the dog-walk when running at high speeds, navigate through the weaves without hitting the poles, and run through tunnels, that eventually can be smaller than the dog itself depending on the breed [19,20]. To prevent injuries and perform at a competition level, agility dogs receive a lot of proprioceptive training, strengthening and balance training, which might lead to a better sense of body-awareness [21–23]. This is also supported by several studies that have shown that highly trained dogs may outperform untrained pet dogs in problems solving tasks [24,25]. An agility-training program for dogs incorporates a variety of obstacles to challenge their physical abilities, coordination, and responsiveness while keeping the sessions engaging and fun. In the main training phase obstacles like tunnels, weave poles, jumps, A-frames, dog walks, and seesaws are introduced. Tunnels are flexible tubes that dogs run through, starting with straight ones and

progressing to curved ones. Depending on the size of the dogs, they have to adjust (crouch down) to fit through. Weave poles require the dog to zigzag through upright poles, while jumps, such as single-bar or tire jumps, can be adjusted in height based on the dog's size and skill level. A-frames and dog walks help improve balance and confidence on elevated surfaces, while seesaws teach dogs to handle shifting surfaces. Positive reinforcement, such as treats, toys, or verbal praise, is essential to encourage the dog and maintain focus, while directional commands like "left," "right," or "jump" improve communication. Sessions should usually be fun, kept short to avoid fatigue, and performed 2–3 times per week, with adjustments based on the dog's fitness level, age, and experience.

In the current study, we aimed to assess if dogs would be a suitable model for a systematic approach to investigating body- and self-awareness in an ecologically relevant context by not only testing the body size perception, but also investigating whether pet dogs are able to assess the shape of their body (regardless of leg length) and whether they understand the solidity of barriers they could move through. Previous studies [1,10] used a single gap paradigm, arguing that this prevents the animals to simply choose a "more convenient" option. We argue that having two gaps is a better approach because to choose a "more convenient" option, the subject still has to have a frame of reference and therefore knowledge of their own size to choose correctly, while when only one choice is available, choosing to go through the gaps may be a matter of motivation to reach the owner who was on the other side of the gap instead of body size perception. Four other studies on snakes, rats, ferrets, and crows respectively [12–15] used a similar idea in a three-choice paradigm. This approach allows for simultaneous comparison of all three sizes (too small, medium, and large) presented in the study. However, there are several issues that arise from this approach. The main issue is the distance between the subject and the openings. The openings are either equidistant from the subject or from the reward, which makes one option more favourable. Indeed, three of those studies [13–15] reported a bias, as ferrets and crows preferred the central opening, while the rats avoided it. This issue alone makes this approach problematic, as we are no longer studying only the preference for the size or shape of the opening, but a combined preference for the location of the opening and the size and shape. Second issue would be the number of combinations needed to test all of the possible combinations of openings presented. Finally, third issue would be the statistical analysis, as with the increase of combination there is an increase of "conditions" that have to be tested separately (either as different models or as different levels of a single variable). Having three option does not allow us to tease apart which opening prompted the animal to make a specific choice and we would have to consider each opening as a separate effect that interact with each other in a single trial. This would render the interpretation very difficult if not impossible.

Accordingly, we used a two-choice paradigm where animals could decide through which opening to pass to get to a reward on the other side. In the first experiment, the openings varied in shape (horizontal, vertical, and circular) and size, while in the second experiment the openings were covered with materials of different solidity (plastic plant, paper, and wood). We also assessed the role of experience in recognizing one's size and shape, and assessing object solidity by comparing pet dogs that have received agility training with pet dogs without agility or any other official (e.g., rescue dogs) training. Given the previous studies on dog body size and shape perception, we would expect dogs to perform well in the two-door paradigm and approach, attempt to walk through, and finally walk through the more appropriate opening more often, i.e., prefer a larger opening over smaller ones and prefer a circular opening over vertical and horizontal openings as it requires least body adjustments, while avoiding the horizontal openings as more body adjustments are needed to walk through them (Experiment 1). We also expected the latency to reach the reward to be smaller the bigger the openings presented. However, if the previous results were not due to body size awareness but rather due to the motivation of the dogs to reach their owner on the other side of the gap, we would expect the dogs to choose at random. We expected that dogs do have an understanding of the solidity of the objects around them and will approach, attempt to walk through, and finally walk through a softer barrier (i.e., the barrier that is easier to tear or push aside) more often. Finally, agility training should greatly improve the dogs' body self-awareness and therefore the dogs with agility training should outperform untrained dogs in all trials.

## Materials and methods

### Ethical statement

Participation was voluntary and the procedure itself was non-invasive and short (~40minutes). The study was discussed and approved by the institutional ethics and animal welfare committee of the University of Veterinary Medicine Vienna in accordance with internal Good Scientific Practice guidelines and national legislation (approval number ETK-140/09/2023). Written consent by the dog owners was obtained prior to the data collection.

### General methods

**Subjects.** A total of 59 pet dogs of different breeds and experience with agility training were tested. We planned to test each dog in two experiments. However, if owners were not able to attend on two separate occasions, their dogs participated in only one of the experiments (41 dogs participated in both experiments, out of which 22 trained and 19 untrained) (see below for separate detailed description of participants in experiment 1 and experiment 2). The time interval between experiments was between 4 and 46 days, depending on the owner availability. A dog was considered agility-trained if they regularly participated in agility training and competed at the A1 level (lowest category to start in agility competitions in Austria). A dog was considered untrained if they had no official training (e.g., no agility, rescue or similar training that would improve their body self-awareness). All owners were recruited using the Clever Dog Lab database or via social media and participated voluntarily. Prior to data collection, the owners had to give written consent.

### Experimental Set-up

The experiment was conducted in the testing room (size: 6m x 7m) of the Clever Dog Lab at the University of Veterinary Medicine, Vienna. A small compartment (1.5m x 3m), containing a food bowl, was separated using fences. There were two openings in the front fences, while the side fences were opaque to minimize distractions. The owner was seated against a wall between the two entrance doors on the opposite side of the room, equidistant to the fence openings. The room was equipped with three cameras, one mounted on the wall above the food reward, one above the owner (model: Panasonic HC-V777), and one centrally on the ceiling (model: AXIS M3058 PLVE) (Fig 1).

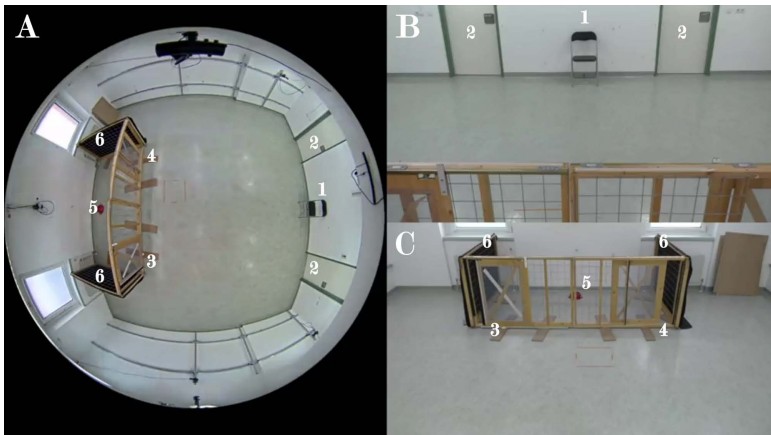

**Fig 1. Experimental setup as seen through (A) the ceiling camera, (B) the camera mounted above the food reward, and (C) the camera mounted above the owner's seat.** The owner entered through one of the two doors (2) and sat on the chair (1). The subject had to choose between the two openings (3 or 4) to reach the reward located in the red bowl (5). Two opaque fences (6) were placed on the sides of the fenced compartment to minimize distractions.

## General procedure

Before each trial, the experimenter placed a food bowl baited with food behind the fences and left the room. The owner then entered the room through one of the two doors with the dog leashed. The entrance door side was randomized and counterbalanced across session. The owner took a seat on the chair that was located between the entrance doors and placed against the wall. The owner then put on a blindfold. The experimenter was positioned outside of the room, monitoring the experiment via live feed from the cameras. The experimenter signalled to the owner to unleash the dog by knocking on the wall from the outside of the room. Once unleashed, the dog was free to roam the room. Since all dogs that were recruited took part in a previous experiment in Clever Dog Lab, they did not require special training to expect and search for food within the testing room. Each trial started when the dog was unleashed and ended when the dog ate the food from the food bowl located in the fenced compartment, or after 1 minute. Once the trial ended, the experimenter knocked on the wall once again to signal to the owner to remove the blindfold, leash the dog, and walk out of the room. Once both the owner and the dog were outside of the room, the experimenter entered the room and baited the food bowl again.

## Training phase

The training phase was carried out to make sure that the animals were comfortable walking through both openings and reaching the reward. Only one opening was open at the time (100 cm x 70 cm), while the other was covered with a wooden plank. The side of the available opening was randomized and counterbalanced across the session. Each subject received 6 training trials in a single session. The subjects had to go through an opening 6 out of 6 times to move on to the test phase. All dogs passed the training criterion in a single session.

## Test phase

In the test phase both openings were covered by different barriers (see below for the respective experiments). Each animal was tested in a single session consisting of 18 trials per experiment. The trial order was randomized and the position of each barrier was counterbalanced across the trials. If the subject stopped participating, a short break of 3 minutes was introduced to motivate the dog to continue participating.

## Video coding

Video coding was performed using Loopy (http://loopb.io, loopbio gmbh, Vienna, Austria). We coded the first side approached, first side attempted, and the side they actually walked through, along with the latency to approach and the latency to reach the reward. First approach was defined as the first time the animal was within one body length radius from an opening. First attempt was defined as the first time the animal tried to walk through the barrier by either putting a paw on the fence frame or by putting the snout against or through the opening. Latency to approach was counted from the moment the dog was released until the subject was within one body length from one of the openings. Latency to reach the reward was counted from the moment the subject was within one body length from one of the openings until the subject reached the red bowl in the fenced compartment. The side to walk through (choice) was simply defined as the opening the animals walked through to reach the reward.

## Statistical analysis

Analyses were carried out in R (version 2023.12.1.402; [26]). Models were fitted using lme4 [27] and survival [28] packages. Significance was inferred via likelihood ratio tests, either by using the drop1 function or by carrying out a full-null comparison (chisquare). All models included training (agility trained or untrained), combination number (used to identify which two choices are presented in each trial; see below for respective experiments), and the interaction between the two as the fixed effects. Each model included sex, z-transformed age (in months), and z-transformed age

squared (in months). Additionally, for Experiment 2 we also included the z-transformed trial number. Trial number was not included for Experiment 1 as the order of combinations presented was randomized and each trial contained a different opening, which would make the interpretation of the effect of trial impossible.We z-transformed trial number and age to a mean of zero and standard deviation of one to enhance model interpretability and facilitate model convergence [29]. We included the subject ID as the random intercept and added random slopes when appropriate. We first checked if there was an overall effect by running a full-null comparison, where the null model included only the intercept (and slope if applicable). If there was an effect, we then determined the effect of the interaction using the drop1 function. If the interaction was not significant, we re-fitted a reduced model without the interaction. The probabilities of choosing the correct choice were calculated using the emmeans package [30]. Finally, we fitted separate models for trained and untrained subjects including only the combination number and random intercept to determine whether the subjects performed differently from chance.

## Experiment 1: Shape and size

### Subjects

A total of 49 pet dogs (29 females and 20 males; mean age: 72 months, range: 14–140) participated in this experiment; 25 pet dogs had previous agility training, while 24 pet dogs had no training.

### Procedure

The openings were covered with differently sized and shaped barriers adapted to the size of the subject. Three rough measurements of each animal were taken using a measuring tape to determine the size of the opening for each animal (see Table 1 below): height at withers (HW), body height (BH), and width of chest (CW) (Fig 2). The horizontal and vertical opening came in three different sizes: a large size that was more than big enough for the subject to walk through it, a medium size that was exactly the size the animal needed to walk through, and a small size that was too small for the animal to fit. The change in size was achieved by either sliding the door on the fence left or right

**Table 1. Formulas used to calculate the different sizes of horizontal, vertical, and circular openings for each animal, using their respective measurements of height at withers (HW), width of chest (CW), and height of body when animal is lying down (BH).**

|  | Horizontal | Vertical | Circular (Ø)* |
|---|---|---|---|
| Large enough | HW * the door width | (1.5 * CW) * the door height | BH |
| Medium size | (2*HW/3) * the door width | CW * the door height | 2BH/3 |
| Too small | (1*HW/3) * the door width | (0.5*CW) * the door height | / |

*Rounded up to the closest multiple of 5.

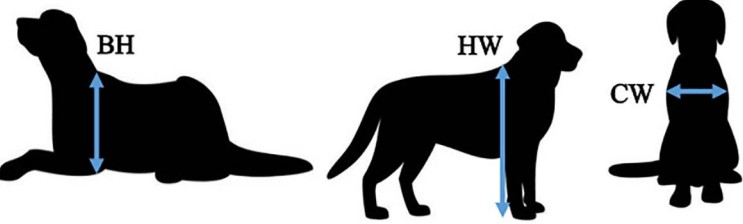

**Fig 2. Measurements taken for each subject: height at withers (HW), width of chest (CW), and height of body when animal is lying down (BH).**

(vertical opening), or by placing the wooden panel higher or lower over the opening (horizontal opening). A circular opening came only in large and medium size. The change in size for circular opening was achieved by attaching panels with different diameters over the circular opening (Fig 3). The panels came in increments of 5 cm, so the size calculated by the formula for the circular opening (Table 1) was rounded up to the nearest multiple of 5. Each subject was presented with 18 different combinations of the openings, seeing each combination only once. Out of the 18 combinations, 4 combinations were used to assess subjects' shape perception by presenting different shapes of the same size (Large or Medium). To assess subjects' size perception, 6 combinations that presented them with different sizes of the same shape (3 for horizontal and 3 for vertical) were used, while the remaining 10 combinations were used to assess the relationship between shape and size by presenting openings that differed in both (for the list of combinations see Table 2).

## Experiment 1: Statistical analysis

For each combination, we defined a correct choice. We first assessed the first 10 combinations that contained only the effect of shape or of size, but not both. Here we defined the Circular opening as always being the correct choice, as it should require the least amount of body adjustment, and Vertical as correct over Horizontal, as it requires the most body adjustment to walk through. Similarly, large openings were always considered the correct choice, while the small openings were always incorrect. We grouped the combinations into four comparison groups (Table 2): Shape1, Shape 2, Size Horizontal, and Size Vertical. For each comparison group, we fitted three Generalized Linear Mixed models (GLMM) with binomial distribution; one model for side that was first approached, one for the side that the subject first attempted to walk through, and one for the actual opening the subject went through. We defined outcome of the trial (coded as 1 or 0) as the response variable. We also fitted a Mixed Effect Cox Model (COXME) to conduct a survival analysis for latencies to approach (from the moment the subject was unleashed until the subject was within one body length from one of the openings) and latencies to reach the reward (from the moment the subject was within one body length from one of the openings until the subject reached the red bowl in the fenced compartment). We then calculated number of times subjects attempted to walk through the same opening they first approached, and number of times subjects chose to walk through the same opening they first attempted. We then ran a binomial test to test if they did so different from chance. Once we ran the analysis on the combinations that had only the effect of shape or of size, but not both, we determined the actual preference of our subjects (i.e., we redefined the correct choices according to our data to make it easier to read the

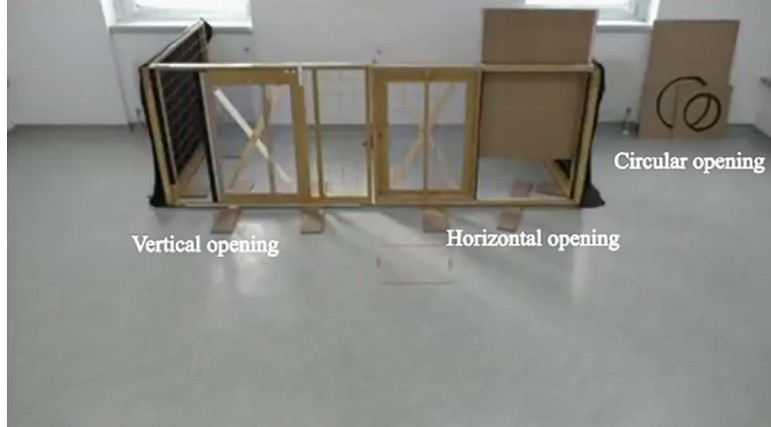

**Fig 3. Apparatus with Vertical and Horizontal barrier in place (Circular openings are visible to the far right).**

**Table 2. List of combinations presented in a single session. The order of the combinations presented was randomised.**

| Combination number | The shape and size of the opening presented | | Group for statistical analysis |
|---|---|---|---|
| 1 | Horizontal large enough | Vertical large enough | Comparison 1:<br>Effect of shape 1 |
| 2 | Horizontal medium size | Vertical medium size | |
| 3 | Horizontal large enough | Circular large enough | Comparison 2:<br>Effect of shape 2 |
| 4 | Vertical large enough | Circular large enough | |
| 5 | Horizontal large enough | Horizontal medium size | Comparison 3:<br>Effect of size horizontal |
| 6 | Horizontal large enough | Horizontal too small | |
| 7 | Horizontal medium size | Horizontal too small | |
| 8 | Vertical large enough | Vertical medium size | Comparison 4:<br>Effect of size vertical |
| 9 | Vertical large enough | Vertical too small | |
| 10 | Vertical medium size | Vertical too small | |
| 11 | Horizontal large enough | Circular medium size | Mixed effect |
| 12 | Vertical large enough | Circular medium size | Mixed effect |
| 13 | Horizontal large enough | Vertical medium size | Mixed effect |
| 14 | Horizontal large enough | Vertical too small | Mixed effect |
| 15 | Vertical large enough | Horizontal medium size | Mixed effect |
| 16 | Vertical large enough | Horizontal too small | Mixed effect |
| 17 | Horizontal medium | Vertical too small | Mixed effect |
| 18 | Vertical medium | Horizontal too small | Mixed effect |

Notes: The position of each barrier was randomized across trials.

results). We then ran two two-tailed binomial test (p = 0.5) for each remaining combination, one for each group of training, to assess whether the shape or the size of the openings are more salient to the animals. Finally, we adjusted the p-values for multiple comparison using the Bonferroni method.

## Experiment 1: Results

We did not detect a significant effect of the interaction between training and combination number in any of our models (p > 0.05), accordingly the results presented here are from the reduced models (not including the interaction).

### Comparison 1: Effect of shape 1 (horizontal vs vertical opening)

The training and combination number did not affect the choice of the side to first approach (Full-null comparison: p = 0.6389), to first attempt (Full-null comparison: p = 0.1459), or to go through (Full-null comparison: p = 0.2185). Untrained dogs performed at chance, while trained dogs approached, attempted to go through, and walked through the horizontal opening significantly more than the vertical one (Table 3, Fig 4).

**Table 3. GLM model outputs for Comparison 1: Effect of shape 1 (combination 1 and 2 from Table 2).**

| Training | First approached | | | First attempted | | | Walked through | | |
|---|---|---|---|---|---|---|---|---|---|
| | p-value | LCL | UCL | p-value | LCL | UCL | p-value | LCL | UCL |
| Trained | 0.0140* | 0.132 | 0.447 | 0.0025* | 0.096 | 0.383 | <0.01* | 0.087 | 0.366 |
| Untrained | 0.2317 | 0.195 | 0.585 | 0.1847 | 0.183 | 0.571 | 0.2821 | 0.204 | 0.596 |

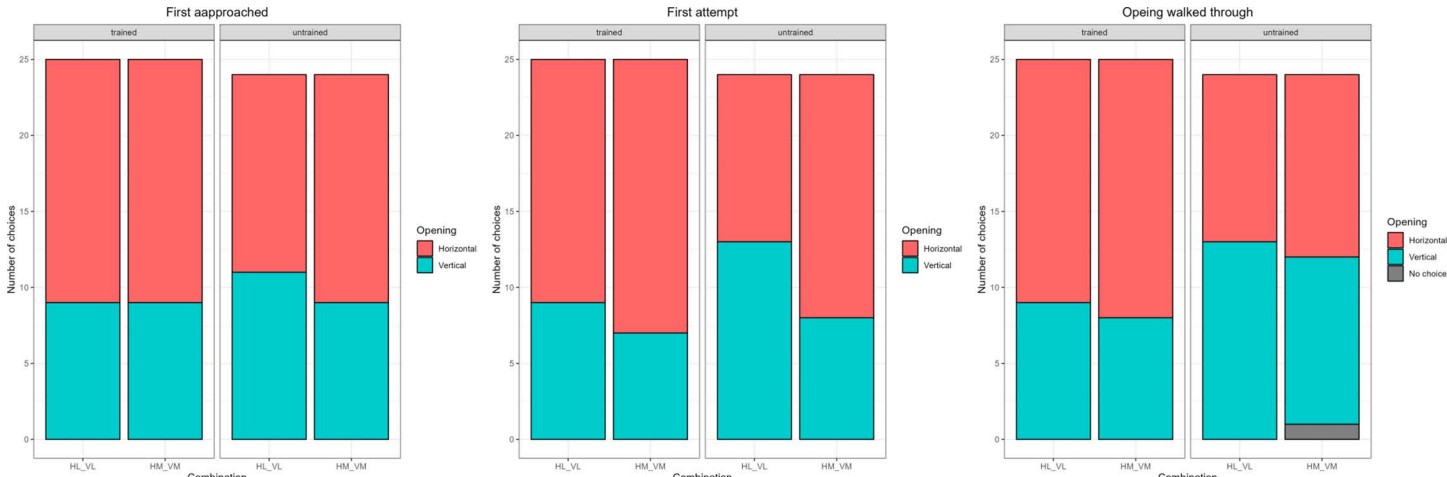

**Fig 4. Barplot showing the number of times each opening was first approached, attempted, and walked through by each subject group in Comparison 1: Effect of shape 1 (horizontal vs vertical opening).** The letters at the bottom represent the combination of openings presented separated by an underscore: first letter represents shape (H for horizontal and V for vertical), while the second letter represents size (L for large and M for medium).

## Comparison 2: Effect of shape 2 (Circular vs Horizontal or Vertical opening)

The training and combination number did not affect the choice of the side to approach (Full-null comparison: p = 0.5394), to attempt (Full-null comparison: p = 0.2928), or to go through (Full-null comparison: p = 0.4175). Both groups performed below chance, i.e., they avoided the circular opening (Table 4, Fig 5).

## Comparison 3: Effect of horizontal size (Horizontal large, Horizontal medium, and Horizontal small)

The training and combination number did not affect the choice of the side to approach (Full-null comparison: p = 0.2815). Both groups approached at chance (Table 5). Training did not affect the choice of the side that was first attempted (p = 0.8642), but there was a significant difference between the combinations. The subjects performed at chance in combination 5 (Large vs Medium; probability to attempt to go through the correct side = 40.1%), while the other two combinations were solved significantly above chance (probability to attempt to go through the correct side in combination 6 (Large vs Small; 75%; p = <0.01) and in combination 7 (Medium vs Small; 96.6%; p < 0.01). For the side chosen to pass through, the model only included combination 5 (Large vs Medium), as it was impossible in combinations 6 and 7 to go through the small opening since the animal could not fit, and thus the outcome could either be correct or no choice. The training did not affect the choice of the side to go through (Full-null comparison: p = 0.779). Both groups walked through the larger opening more often than the medium sized opening (Table 5, Fig 6). The square of age had a significant effect on the choice of the side subjects first attempted (p = 0.0171).

**Table 4. GLM model outputs for Comparison 2: Effect of shape 2 (combination 3 and 4 from Table 2).**

| Training | First approached | | | First attempted | | | Walked through | | |
|---|---|---|---|---|---|---|---|---|---|
| | p-value | LCL | UCL | p-value | LCL | UCL | p-value | LCL | UCL |
| Trained | 0.0126* | 0.125 | 0.442 | <0.01* | 0.069 | 0.348 | <0.01* | 0.028 | 0.27 |
| Untrained | 0.5073 | 0.222 | 0.648 | 0.0184* | <0.001 | 0.241 | 0.0133* | <0.001 | 0.173 |

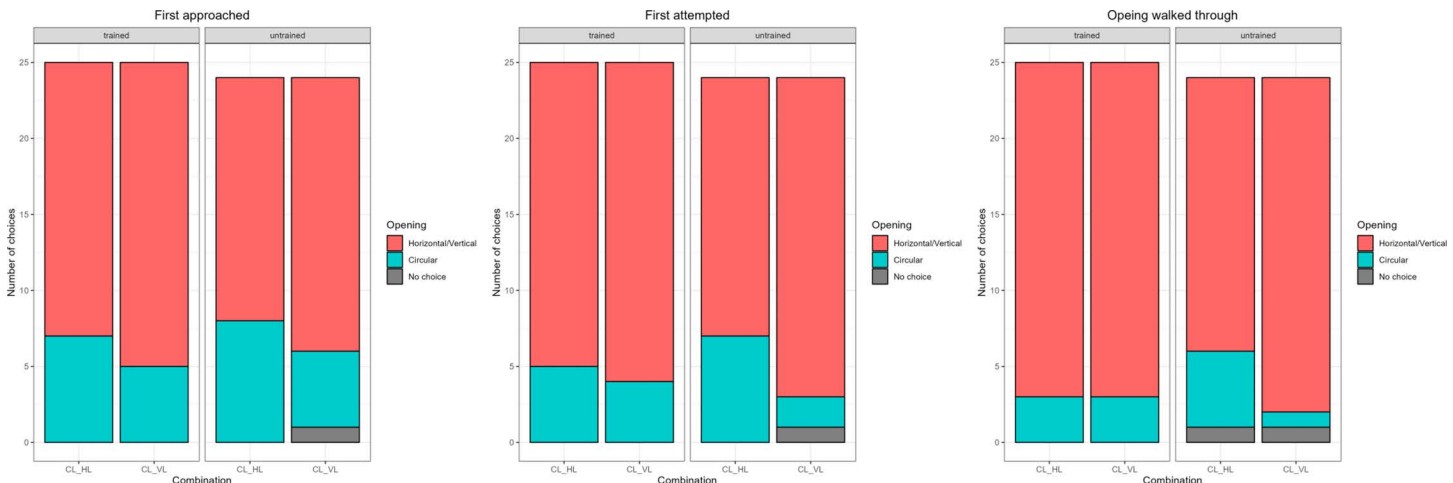

**Fig 5. Barplot showing the number of times each opening was first approached, attempted, and walked through by each subject group in Comparison 2: Effect of shape 2 (Circular vs Horizontal or Vertical opening).** The letters at the bottom represent the combination of openings presented separated by an underscore: first letter represents shape (H for horizontal, V for vertical, and C for circular), while the second letter represents size (L for large).

**Table 5. GLM model outputs for Comparison 3: Effect of horizontal size (combination 5, 6, and 7 from Table 2).**

| Training | First approached | | | First attempted | | | Walked through (combination 5 only) | | |
|---|---|---|---|---|---|---|---|---|---|
| | p-value | LCL | UCL | p-value | LCL | UCL | p-value | LCL | UCL |
| Trained | 0.3060 | 0.425 | 0.725 | <0.0122* | 0.549 | 0.836 | <0.01* | 0.677 | 0.918 |
| Untrained | 0.6774 | 0.361 | 0.706 | <0.0522' | 0.499 | 0.844 | 0.0219* | 0.532 | 0.894 |

## Comparison 4: Effect of vertical size (Vertical large, Vertical medium, and Vertical small)

The training did not affect the choice of the side to approach (p = 0.4255), to attempt (p = 0.7163), or to go through (p = 0.607). However, there was a significant effect of combination number. The subjects first approached the correct side more often in combination 9 (Large vs Small; 79.6%; p < 0.01), while they approached at chance in the other two combinations (combination 8, Large vs Medium; 49.1%, combination 10, Medium vs Small; 43.7%). They first attempted the correct side significantly above chance in combinations 9 and 10 (probability to attempt the correct side in combination 9 = 87.2%; p < 0.01, combination 10 = 80.7%, p = 0.0078), while they attempted at random in the combination 8 (Large vs Medium, 52%). Finally, for the side to walk through, the model only included combination 8 (Large vs Medium), as it was impossible to pass through the small one (as the animal could not fit) in combinations 6 and 7, and the outcome could either be correct or no choice. Both trained and untrained dogs preferred the larger opening over the medium opening (Table 6, Fig 7).

## Mixed effect combinations

According to results presented, we expected that the preferences will remain the same for the mixed effect combination (number 11–18, Table 2), i.e., that the untrained animals will not show a preference for the Horizontal over the Vertical openings. Furthermore, we expected that the Circular and small openings would be avoided. Finally, we expected the large opening to be preferred over a medium one. The animals walked through the correct opening in

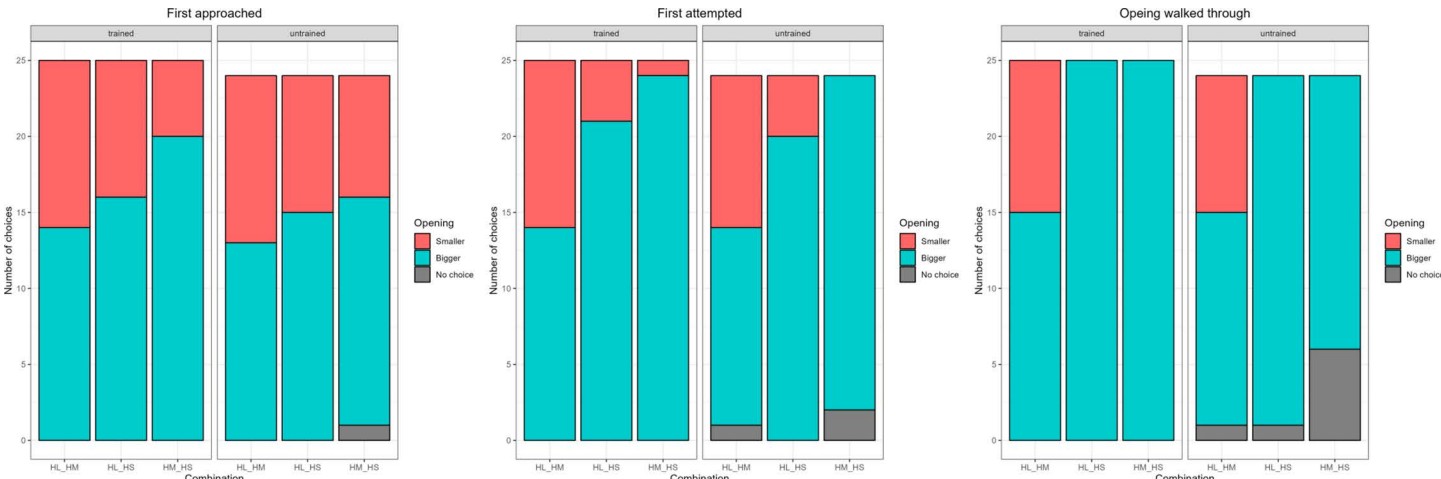

**Fig 6. Barplot showing the number of times each opening was first approached, attempted, and walked through by each subject group in Comparison 3: Effect of horizontal size (Horizontal large, Horizontal medium, and Horizontal small).** The letters at the bottom represent the combination of openings presented separated by an underscore: first letter represents shape (H for horizontal), while the second letter represents size (L for large, M for medium, and S for small).

**Table 6. GLM model outputs for Comparison 4: Effect of vertical size (combination 8, 9, and 10 from Table 2).**

| Training | First approached | | | First attempted | | | Walked through (combination 8 only) | | |
|---|---|---|---|---|---|---|---|---|---|
| | p-value | LCL | UCL | p-value | LCL | UCL | p-value | LCL | UCL |
| Trained | 0.2161 | 0.443 | 0.733 | <0.01* | 0.593 | 0.853 | <0.01* | 0.639 | 0.892 |
| Untrained | 0.2566 | 0.428 | 0.762 | 0.0201* | 0.55 | 0.9 | <0.01* | 0.699 | 0.943 |

all combinations except for combinations 13 (Horizontal large vs Vertical medium) and 15 (Vertical large vs Horizontal medium) (S1 Table).

## Latency to approach

We found no significant effect in comparisons 1 (Full-null comparison (before excluding the interaction): p=0.0667), 2 (Full-null comparison (before excluding the interaction): p=0.2212), and 4 (Full-null comparison (before excluding the interaction): p=0.2905). In comparison 3 (Horizontal Large, Medium, and small) training (p=0.0240) and sex (p=0.0389) had a significant effect on latency to approach. Untrained (hazard ratio=0.4125) dogs were slower, while male dogs (hazard ratio=2.1686) were faster to approach the apparatus.

## Latency to reach reward

Combination number significantly affected the latency to reach the reward in comparisons 1, 3, and 4, while we detected a trend in comparison 2 (Table 7). The effect of training was significant only in comparisons 2, 3, and 4 (Table 7). Overall, trained dogs reached the reward faster and had a higher probability of walking through an opening within the time limit. The subjects took longer to reach the reward and had a lower probability to walk through an opening within the time limit if the presented barriers were smaller (S1 Fig). Z-transformed age had a significant effect in comparisons 1 (p<0.01) and 4 (p=0.0273). The dogs with age in the extremes of our sample (younger and older dogs) were slower, with the effect being stronger for older dogs (S2 Fig).

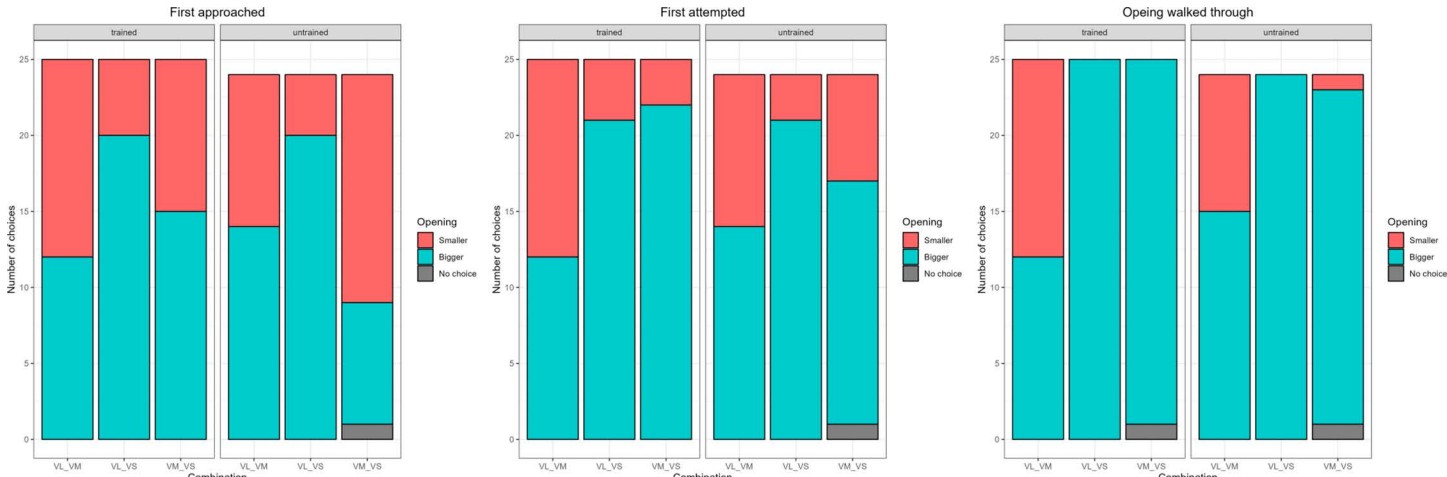

**Fig 7. Barplot showing the number of times each opening was first approached, attempted, and walked through by each subject group in Comparison 4: Effect of vertical size (Vertical large, Vertical medium, and vertical small).** The letters at the bottom represent the combination of openings presented separated by an underscore: first letter represents shape (V for vertical), while the second letter represents size (L for large, M for medium, and S for small).

**Table 7. Coxme model outputs for latencies in Experiment 1. For each comparison (Table 2) we presented the comparison of groups (first group vs second group), hazard ratio, and p-value (an apostrophe (') indicates a trend, while an asterisk (*) indicates significance).**

| | Comparison 1: effect of shape 1 | | | Comparison 2: effect of shape 2 | | | Comparison 3: effect of size horizontal | | | Comparison 4: effect of size vertical | | |
|---|---|---|---|---|---|---|---|---|---|---|---|---|
| | Compari-son | Hazard ratio | p-value | Compari-son | Hazard ratio | p-value | Compari-son | Hazard ratio | p-value | Compari-son | Hazard ratio | p-value |
| Effect of combination number | 1 vs 2 | 0.3813 | <0.01* | 3 vs 4 | 1.5677 | 0.0551' | 5 vs 6 | 0.5672 | 0.0147* | 8 vs 9 | 0.6251 | 0.0269* |
| | | | | | | | 5 vs 7 | 0.2416 | <0.01* | 8 vs 10 | 0.3228 | <0.01* |
| Effect of training | Trained vs Untrained | 0.4939 | 0.1303 | Trained vs Untrained | 0.4888 | 0.0388* | Trained vs Untrained | 0.5008 | 0.0484* | Trained vs Untrained | 0.6134 | 0.0064* |

### Relation between approach and attempt, and attempt and choice

Our subjects first attempted the opening they first approached significantly above chance (723 out of 871 cases; p < 0.01) and walked through the opening they first attempted significantly above chance (794 out of 853 cases; p < 0.01).

### Experiment 1: Discussion

Our results suggest that animals did correctly assess their size, as they attempted and went through the correct barrier above chance in Comparisons 3 (Horizontal shapes of different size) and 4 (Vertical shapes of different size). Both groups also took longer to walk through an opening when the presented openings were smaller (medium and small size). These results are in line with previous studies [1,10] as dogs took longer to reach the reward when the openings presented were smaller. The agility training did not affect the choice of the animals, but it greatly affected the latencies to reach the reward once the first barrier had been approached. Agility dogs were quicker and were more likely to walk through an opening within the time limit, suggesting that both animal groups had a notion of their size, but agility dogs needed less time to assess the size of the openings in front of them. However, there was no difference in latencies to approach the apparatus in comparison 4 (Vertical openings). Since we showed that most subjects actually approached, attempted, and walked

through the same opening, it would seem that agility trained dogs hesitated less once they approached the opening. The trained dogs were faster to approach the apparatus in comparison 3 (Horizontal openings).

Untrained dogs did not have a preference for the horizontal or vertical shape, while the agility dogs preferred the horizontal opening. This is surprising, since no body adjustment was needed to walk through the vertical openings, while the subjects had to at least duck their head to walk through the horizontal opening. Agility dogs may have more experience with ducking under an obstacle and may have felt more familiar with this shape, which would explain the effect we detected. Taken with the shorter latency to approach when horizontal openings are presented, the most likely explanation is that agility dogs learned how to handle horizontal openings through their training, while vertical openings are usually avoided. This is probably due to most obstacles being horizontal or circular on the agility course, while the closest to a vertical opening that they encounter during training are weave poles, which are flexible and can be pushed aside, while the fences in our experiment were rigid and could not be easily moved. Additionally, several animals hit the brim of the horizontal opening with their back. This suggests that they may not be fully aware of their body's shape, but more investigation is required to draw a conclusion. Unfortunately, the hitting of the opening was not clearly visible on the videos, as this was not the focus of the study, so we could not assess whether previous agility training affected the amount of times each subject hit the brim. However, the subjects showed no aversion or fear of the apparatus after hitting the barrier, which might mean that dogs did not even pay attention to this.

Interestingly, the animals did not have a preference in mixed effect combination where horizontal and vertical openings came in different sizes. Both groups overall avoided circular opening. This is probably due to the fact that to go through a circular opening the subjects had to jump with their hind legs, which poses a risk of hitting the brim and would therefore be a less desirable option. The exact reason why the subjects avoided the circular opening remains unclear. It would therefore be beneficial if future studies also included combinations where circular option is presented with medium and too small vertical and horizontal openings. Although, given that our subjects avoided the circular opening, if it is presented with a too small opening it might be perceived as an impossible choice, which will probably results in subjects not making a choice at all. It would however seem that shape and size are assessed separately, as the untrained animals retained their preferences for shape and size even when presented with mixed combinations of openings (combination number 11–18). This connection (or lack thereof) between size and shape should be studies in greater detail. Agility dogs, however, did not show preference of the horizontal openings in the mixed combinations. This is interesting as it suggests that the perception of shape and size interact with each other, which caused the subjects to choose at random when the openings differed in both shape and size.

Finally, the effect of age on the first opening attempted in Comparison 3 (Horizontal openings) is likely coincidental, as we found no effect of age in other comparisons and no effect of age when approaching or walking through an opening in Comparison 3. Similarly, the effect of sex on latency to approach in comparison 3 is likely a coincidence too. The finding that the younger and older dogs took longer to reach the reward in Comparisons 1 and 4 probably has nothing to do with size or shape perception. Most likely explanation is that younger dogs were more explorative or distracted, while the older dogs simply walked slower. Since all of our subjects were adults, we did not expect age to have any effect on perception of shape and size, as they are already fully-grown and the mental image of their bodies should have already been fully formed. Future studies could focus on the effect of age on size and shape perception to investigate the matter further.

## Experiment 2: Object solidity

### Subjects

A total of 51 pet dogs (28 females and 23 males; mean age: 70 months, range: 15–141) participated in this experiment; 26 pet dogs had agility training, while 25 pet dogs had no training.

## Procedure

Prior to the test phase, each subject received a short familiarization phase during which the animal was free to search for a treat in a wooden box (Fig 4). The wooden box was either made entirely out of wood or had one side made out of one of the other two materials (paper or decorative plastic plant), which had to be pushed or torn to reach the reward. This way, we made sure that each subject had the necessary exposure to the three materials. The box had a removable wooden lid with a latch. In case of the paper and the plant condition, we removed the wooden lid and covered the opening with the other material. The paper was attached using painter tape, while the plastic plant strings were attached to a sewing thread, which was held in place by two knots protruding through the box's frame. The owner with the dog leashed entered and moved to the middle of the room. The experimenter showed the reward to the subject, and then placed it in the box, covering the opening with the appropriate material. The experimenter then placed the box in front of the subject, with the covered opening facing it, and took one step back. The owner then unleashed the dog. If the dog stopped trying to reach the reward and either started exploring the room or looked toward the owner for assistance, the experimenter instructed the owner to encourage the dog to reach the reward. For the condition when the wood was covering the opening, it was impossible for the subject to reach the reward and the trial ended after 1 minute or if the subject showed signs of frustration (e.g., barking or whining). Each subject received 2 trials per material in the familiarization phase. All participants managed to push the plant and tear the paper to reach the reward.

The openings on the front fences were completely covered by barriers made out of different materials (wood, paper, or a decorative plastic plant) (Fig 8). The test was conducted in the same way as in experiment 1. Each dog received 6 trials per possible combination of the materials adding up to a total of 18 trials in a single session. The position of each material was randomized and counterbalanced across trials.

## Experiment 2: Statistical analysis

We defined the correct choice for each of the 3 possible combinations (plant vs paper, plant vs wood, and paper vs wood). Plastic plant was always considered correct, as it required the least amount of force to push aside, while the wood was always considered incorrect, as it was impossible to walk through it. We fitted three Generalized Linear Mixed models (GLMM) with binomial distribution; one model for barrier that was first approached, one for the barrier that the subject first attempted to walk through, and one for the actual opening the subject went through. We defined outcome of the trial (coded as 1 or 0) as the response variable. We also fitted a Mixed Effect Cox Model (COXME) to conduct a survival analysis for latencies to approach (from the moment the subject was unleashed until the subject was within one body length from one of the openings) and latencies to reach the reward (from the moment the subject was within one body length from one of the openings until the subject reached the red bowl in the fenced compartment).

## Experiment 2: Results

We did not detect a significant effect of the interaction between training and combination number (used to identify the barriers presented in the trial) in any of our models (p > 0.05), so the results presented here are from the reduced models (not including the interaction).

Agility training (Table 8) affected the probability to attempt the correct side first and to walk through the correct side, while we detected a trend for approach. Combination number (Table 9) affected the probability to both approach and attempt the correct side first, while the trial number had no effect (effect of z-transformed trial number: approach p-value = 0.9782; attempt p-value = 0.1646) (Fig 9). The model for side chosen to pass through included only combination 1 (plant vs paper), as it was impossible to choose incorrectly in the other two combinations (combination 2: plant or combination 3: paper vs wood). Untrained dogs outperformed trained dogs when approaching, attempting, and choosing the side to walk through (Table 8), while the probability to approach the correct side first was lower for combination 3 (paper vs wood) when compared to combinations 1 (p-value = 0.0148) and 2 (p-value = 0.0069) and the probability to attempt the correct side first was highest for combination 2 (plant vs wood) when compared to combinations 1 (p-value = 0.0001) and 3 (p-value = 0.0224).

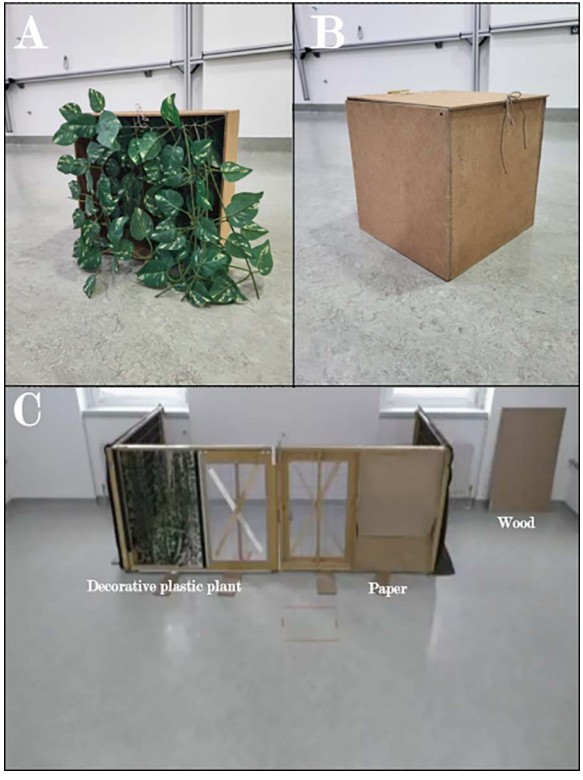

**Fig 8. Apparatus used in Experiment 2.** A: The box used in the familiarization phase, with one side covered by plant. B: The box used in the familiarization phase, completely made out of wood. C: The two openings covered with different materials (left: plant, right: paper, and the wooden plank is visible in the corner of the room for clarification purposes and was not in the room during testing).

**Table 8. GLM model outputs for Experiment 2 (Solidity) for the first side approached, first side they attempted to walk through and the side they actually walked through (combination 1 only) for each training group.**

| Training | First approached | | | | First attempted | | | | Walked through (combination 1 only) | | | |
|---|---|---|---|---|---|---|---|---|---|---|---|---|
| | p-value | probability | LCL | UCL | p-value | probability | LCL | UCL | p-value | probability | LCL | UCL |
| Trained | 0.054* | 50.4% | 0.448 | 0.56 | 0.0253 | 70.5% | 0.632 | 0.768 | 0.0264* | 65.1% | 0.550 | 0.740 |
| Untrained | | 57.9% | 0.516 | 0.639 | | 80.1% | 0.731 | 0.856 | | 78.8% | 0.689 | 0.862 |

**Table 9. GLM model outputs for Experiment 2 (Solidity) for the first side approached and first side they attempted to walk through for each combination of materials.**

| Combination | First approached | | | First attempted | | |
|---|---|---|---|---|---|---|
| | probability | LCL | UCL | probability | LCL | UCL |
| plant vs paper | 57.5% | 0.510 | 0.637 | 68.1% | 0.603 | 0.749 |
| plant vs wood | 58.4% | 0.520 | 0.646 | 82.9% | 0.768 | 0.876 |
| paper vs wood | 46.1% | 0.398 | 0.526 | 73.7% | 0.662 | 0.800 |

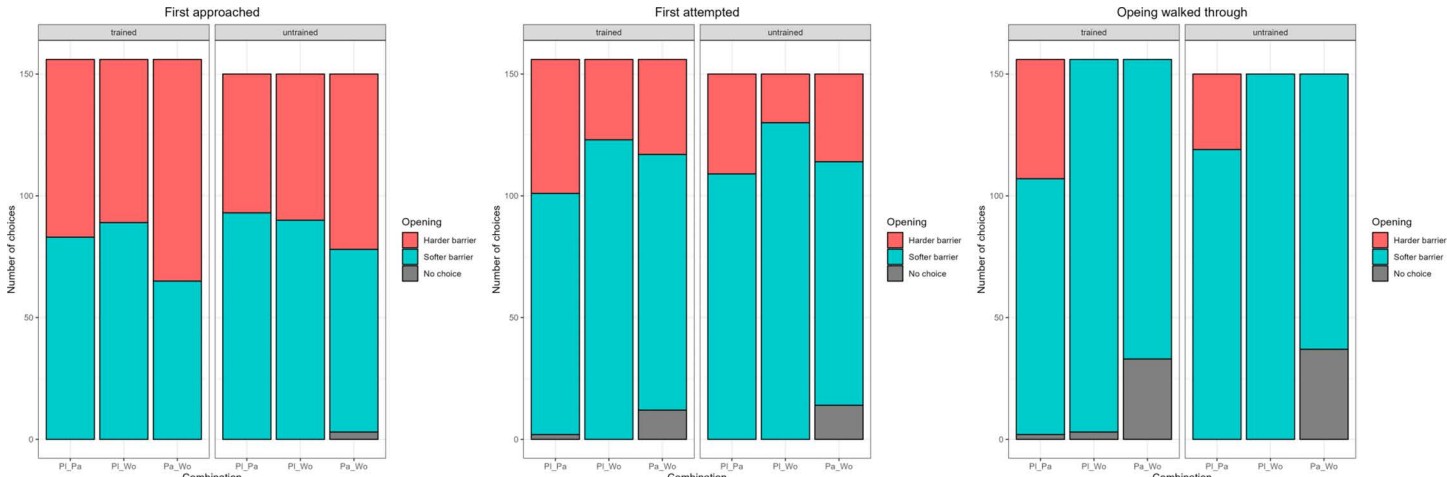

**Fig 9. Barplot showing the number of times each opening was first approached, attempted, and walked through by each subject group in Experiment 2.** The letters at the bottom represent the combination of openings presented separated by an underscore (Pl for plastic plant, Pa for paper, Wo for wood).

## Latency to approach

The training did not have an effect on latency to approach the apparatus. However, there was a significant difference between combinations (Table 10). Subjects approached the apparatus slower in combination 3 (paper vs wood) than in other two combinations.

## Latency to reach reward

The training did not have an effect on latency to reach the reward. However, there was a significant difference between combinations (Table 11). The subjects had the highest probability to make a decision and did so faster in Combination 1,

**Table 10. Coxme model outputs for latencies to approach in Experiment 2: the comparison of groups (first group vs second group), hazard ratio, and p-value (an asterisk (\*) indicates significance).**

|  | Comparison | Hazard ratio | p-value |
|---|---|---|---|
| Effect of training | Trained vs Untrained | 0.9821 | 0.9548 |
| Effect of combination number | 2 vs 3 | 1.37 | <0.01* |
|  | 1 vs 2 | 1.04 | 0.9081 |
|  | 1 vs 3 | 1.32 | <0.01* |

**Table 11. Coxme model outputs for latencies to reach reward in Experiment 2: the comparison of groups (first group vs second group), hazard ratio, and p-value (an apostrophe (') indicates a trend, while an asterisk (\*) indicates significance).**

|  | Comparison | Hazard ratio | p-value |
|---|---|---|---|
| Effect of training | Trained vs Untrained | 0.9313 | 0.5735 |
| Effect of combination number | 2 vs 3 | 3.53 | <0.01* |
|  | 1 vs 2 | 1.45 | <0.01* |
|  | 1 vs 3 | 5.12 | <0.01* |

while the probability was lowest and the subject needed more time to make a decision in Combination 3 (S3 Fig). Z-transformed age also had a significant effect. The dogs with age in the extremes of our sample (younger and older dogs) were slower, with the effect being stronger for older dogs (S2 Fig).

### Relation between approach and attempt, and attempt and choice

Our subjects first attempted the opening they first approached significantly above chance (669 out of 890 cases; p < 0.01) and walked through the opening they first attempted significantly above chance (721 out of 843 cases; p < 0.01).

## Experiment 2: Discussion

Our results suggest that our subjects managed to assess the solidity of materials even after a short interaction with it and could discriminate between them. The untrained dogs outperformed trained dogs when approaching, attempting, and choosing the barrier to walk through, which is surprising. The trained dogs did not have a preference when approaching, while the untrained dogs approached the correct side above chance. Both groups attempted and walked through the correct opening above chance.

Although we detected a significant difference between groups when approaching, the difference in estimated probability to approach the correct first was quite small. It is still surprising that untrained dogs outperform trained dogs, as it would be expected that agility training would improve quick decision-making and that trained dogs would then be able to assess the solidity faster than untrained dogs. There was no significant difference between the groups in latency to approach. Interestingly, we did not detect a significant difference in latency to reach reward between the groups (unlike in Experiment 1). This was also surprising as we expected the agility trained dogs to be faster as speed is an important factor in agility competitions. It is possible that agility training changed how dogs assess the physical feature of the world around them and thus they made a choice once they were close to the barrier, while the untrained dogs may have done so while approaching. This also could have in turn affected the latency to reach the reward. Alternatively, trained dogs might have been more socially inhibited by their owner's presence [31,32] as they have more training experience, and therefore hesitated to "break" a barrier, as this is usually undesired in normal owner-dog interactions. Furthermore, to successfully finish an agility course it is usually expected of the animals not to touch or break any obstacles, meaning that agility trained dogs were specifically trained not to "break" a barrier, which in turn would increase their latency to reach the reward. More experiments would be needed to analyse how training changes the assessment of different physical attributes of other objects in the environment. It might also be beneficial to test the dogs in the absence of their owners, however this may lead to less exploration and activity, which would directly influence their willingness to walk through a barrier [33]. We would expect the dogs to become better at recognizing different materials as they have more experience the older they are. Therefore, the finding that the age had a similar affect as it did in Experiment 1 further supports the idea that this is a result of how dogs behave during testing and not an effect of age on physical cognition. Alternatively, it is possible that with age dogs become worse at telling materials apart. We find this explanation unlikely, but more research would be needed to confirm that statement. Effect of age was not the focus of this study.

The weaker (albeit positive) preference to walk through the correct opening in trained dogs could be explained by them having more experience and encountering more materials in their lives, which would result in a lower perceived difference in solidity between plant and paper. To confirm this interpretation, future studies should include different materials to further research how dogs assess solidity. Finally, the influence of experience should be studied in greater detail by including dogs with different types of training.

## General discussion

Dogs have been suggested as a suitable model to investigate body- and self-awareness in an ecologically relevant context [1,34,35]. Here we expanded previous studies by not only investigating the perception of body size, but also body

shape (in a more general sense) and by exploring animals' ability to understand the solidity of barriers in the environment. In this study, we used a two-choice paradigm to assess dogs' perception of their own size and shape (Experiment 1), and their understanding of solidity of obstacles (Experiment 2). We also investigated the effect of agility training on dogs' body awareness by comparing trained and untrained dogs.

Both groups of dogs avoided the circular opening and preferred large openings to medium and small openings. Untrained dogs had no preference when approaching, while trained dogs did. This suggests that agility dogs made a decision during approach, while untrained dogs made a decision once they reached the apparatus. Untrained dogs had no preference between vertical and horizontal shape. This is in line with the previous results by Lenkei and her colleagues (experiment 3) [1], where they showed that dogs did not rely on shape perception when arriving at the opening. Interestingly, agility trained dogs preferred the horizontal opening to the vertical one. This is probably due to most obstacles being horizontal or circular on the agility course, so through their training agility dogs learned how to handle these obstacles. The closest to a vertical opening that they encounter during training are weave poles, which are flexible and can be pushed aside, while the fences in our experiment were rigid and could not be easily moved. To investigate this further, more studies focusing on animals with different life experience and specific training are required. The agility trained dogs also had shorter latencies to reach the reward. Overall, the latencies were longer the smaller the barriers presented. The exact reason why the subjects avoided the circular opening remains unclear. In Experiment 2, both groups preferred the plant to the paper barrier, but the untrained dogs had a stronger preference than the agility trained dogs. The probability to walk through a barrier within the time limit was higher when the wooden (impassable) barrier was not present. Both groups attempted to walk through the correct side first more often in combination 2 (plant vs wood) when the difference in solidity of barriers was highest, while the preference was the weakest for combination 1 (plant vs paper) when the difference in solidity was the lowest.

The significance of our findings is three fold: we evaluated the two-choice paradigm, we examined dogs' body size, shape, and object solidity perception, and we studied the influence of experience on ecological self. We challenged the statement by Lenkei and her colleagues [1] that dogs would always choose a "more convenient" option if tested in a two-choice paradigm. We argue that a two-choice paradigm is justified, since testing animals with a single opening does not allow us to disentangle the effect of the size of the opening and the motivation of the animal. It is also needed to have a frame of reference to determine what the "more convenient" option would be. Additionally, if there is only one opening present, then that is the only way to reach the reward. When we present the animals with two options, we can directly compare two openings, which provides us with insight into how different shapes and sizes are processed. Furthermore, Lenkei and her colleagues also stated that the latency to approach is a better measurement of body-size perception than the latency to reach reward, as the latency to reach the reward is more driven by the motivation than other factors. Here we found that untrained dogs approached at random which implies that the latency to reach the reward gives us information on hesitation once the animal interacted with the apparatus and came close enough to make a choice.

Our results in Experiment 1 are in line with the existing literature [1,10], as dogs took longer to walk through smaller openings. Although both previous studies demonstrated dogs' flexibility when navigating openings of different size (and shape [1]), the previous studies never directly compared two openings of different size and shape. Here we demonstrated that untrained dogs will not show a preference for shape, as long as the opening is large enough for them to fit and does not require additional body adjustments. As mentioned above, these results are in line with the results reported by Lenkei and her colleagues, therefore although we criticized the single-choice paradigm, we confirmed their findings, which suggests that it is still a valid method. This rises several interesting points. Firstly, it seems that dogs do have an accurate assessment of their body shape and size and can use this information in quick decision-making. Secondly, the preferences for shape and size disappeared in combinations where openings varied in both, which implies that the effects of size and shape interact. This suggests that within one "building block" of ecological self, there may be several different aspects, which carry different information, that are taken together to form a mental image of "self". Importantly, unlike

the building blocks analogy, these aspects may not be hierarchical. This expands the idea that ecological self can be divided into building blocks which vary across species and ecological niches [3,4], as some building blocks may be further divided into smaller units, which can and should be studied both separately and together. It would therefore be interesting to design an experiment where multiple physical attributes of a barrier have to be taken into an account to test whether smaller units within a building block have different importance in determining a preference.

Interestingly, we found no effect of training on size perception, but the agility-trained dogs had a preference for horizontal openings This is not surprising since most obstacles used in agility training are horizontal, meaning agility dogs had more experience with this shape and already knew how to handle such obstacles at high speed. Adding to this, although both groups correctly assessed the solidity of the barriers covering the openings, the untrained dogs performed better in Experiment 2. There are two possible explanations for this. Either the agility-trained dogs gave more importance to other information from their environment and did not focus on solidity of the barrier in front of them. This is unlikely, since they still performed above chance. It is much more likely that agility-trained dogs had experience with more materials and therefore the perceived difference between solidities was smaller. Therefore, experience may provide a higher resolution or a bigger range of the information received from the environment and in turn does not change how animals assess the physical attributes of objects around them, but how this information is processed. To further investigate how experience affects the processing of information on physical attributes from their environment, it would be important to test dogs with different specific training (e.g., rescue, guide, police dogs, etc.).

Given that we find evidence of ecological self across taxa (elephants [8], ferrets [15], rats [13], snakes [12], budgerigars [11], crows [14], hermit crabs [17], cats [16], and dogs [1,9,10]), it would seem that the more important criterion for a suitable model to systematically investigate different building blocks of self might be 1) availability of a large enough sample size and 2) the ease of working with the chosen species, rather than cognitive abilities. This is further supported by the relatively small sample sizes of some of the previous studies (6 [13–15], 7 [11], 12 [8], 20 [12]). It would seem that the highest sample size achieved was either using a study species that appears in great numbers in the wild (hermit crabs, n = 40 [17]) and by testing pet animals (cats: 30 [16]; dogs: 39 [1], 44 [10], 54 [9], and 59 in the current study). Dogs are the most common pet animal and they are already used to spending time outside of their homes, which makes them suitable for both at home and lab testing. Furthermore, since there are several well-defined specific training programs for dogs (e.g., agility, rescue, and military), they are the perfect model to test the influence of different life experiences on the perception of self. In addition, testing free-ranging dogs can be another venue of approach. Finally, we could also compare dogs with their closest living relative, wolves, to evaluate the effect of domestication on the perception of self. Taken together, dogs are a prime candidate for a model species to investigate the building blocks of self.

Dogs seem to be a suitable model for a systematic approach to investigating body- and self-awareness in an ecologically relevant context. We demonstrated that dogs do have a notion of their body-size, as they preferred the bigger openings more often. We also demonstrated that dogs showed understanding of the solidity of barriers as they attempted and walked through a less solid barrier more often. The agility training did not influence the size perception. However, the agility training changed shape perception and shortened the time needed for the dogs to make a decision when approaching the openings of different size and shape, but increased the time when approaching openings covered with different materials. Our results show that specific training may influence how information on different physical attributes may be processed. This further confirms that "building blocks" are highly context specific and impacted by life experience. We require more studies following similar procedures testing for more "building blocks" of self-awareness in dogs, and the underlying aspects of each building block, to create a systematic overview of how each of these cognitive abilities contributes to the perception of self and how do they manifest in an ecologically relevant contexts. To get a broader picture of the decision making process, in addition to side chosen and latencies, further studies could also analyse hesitation behaviour and bodily adjustment. We also require more studies on physical attributes of a body and objects in the environment in

different context and including animals with different life experience to learn how these perceptions are formed and what circumstances improve (or deteriorate) the underlying cognitive processes.

## Supporting information

**S1 Table. p-values adjusted for multiple comparison using the Bonferroni method.**
(DOCX)

**S1 Fig. Survival plots for latency to reach reward in a) Comparison 1: Effect of shape 1 (horizontal vs vertical, b) Comparison 2: Effect of shape 2 (circular vs horizontal or vertical, c) Comparison 3: Effect of size horizontal (horizontal large, medium, and small), d) Comparison 4: Effect of size vertical (vertical large, medium, and small).** First letter of the names in the legend indicate shape (H-horizontal, V-vertical, C-circular) and the second indicates size (L-large, M-medium, or S-small).
(PNG)

**S2 Fig. Plots showing the effect of age in Comparisons 1 and 4 in Experiment 1 and in Experiment 2.**
(PNG)

**S3 Fig. Survival plot for latency to reach reward in Experiment 2.** The names in the legend are first two letters of the material covering the material (Pl is short for plastic plant, Pa for paper, and Wo for wood).
(PNG)

**S1 File. Script used to run the analysis.** This is the original R script used for statistical analysis for the original submission of the manuscript.
(R)

**S2 File. Script used to run the analysis.** This is the R script used for statistical analysis containing all the necessary changes for the first round of review.
(R)

**S3 File. Script used to run the analysis.** This is the final R script used for statistical analysis containing all the necessary changes for the second round of review. The results reported above were obtained using this script.
(R)

**S4 File. Readme file for the R scripts.** A file containing a few clarifications to help a reader understand the script provided (File S1, S2, and S3).
(TXT)

**S5 File. Full dataset used for the analysis.** The full dataset collected and analysed for this study.
(CSV)

## Author contributions

**Conceptualization:** Dario Staric, Lea Arnauer, Sarah Marshall-Pescini, Friederike Range.

**Data curation:** Dario Staric, Lea Arnauer.

**Formal analysis:** Dario Staric, Lea Arnauer.

**Funding acquisition:** Sarah Marshall-Pescini, Friederike Range.

**Investigation:** Dario Staric, Lea Arnauer.

**Methodology:** Dario Staric, Lea Arnauer.

**Project administration:** Dario Staric.

**Supervision:** Dario Staric, Sarah Marshall-Pescini, Friederike Range.

**Validation:** Dario Staric, Lea Arnauer.

**Visualization:** Dario Staric.

**Writing – original draft:** Dario Staric, Lea Arnauer.

**Writing – review & editing:** Dario Staric, Sarah Marshall-Pescini, Friederike Range.

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
