## [Decision Letter · Decision Letter 0]

2 Jul 2025

Dear Dr. Staric,

Thank you for submitting your manuscript to PLOS ONE. After careful consideration, we feel that it has merit but does not fully meet PLOS ONE’s publication criteria as it currently stands. Therefore, we invite you to submit a revised version of the manuscript that addresses the points raised during the review process. 

After reviewing the feedback from three qualified reviewers, I have decided to request major revisions to your work. All three reviewers agree that your study is novel and makes significant contributions to the field of dog cognition. However, they have identified several major limitations in the methodology that must be addressed before your article can be considered for publication. Please note that while all three reviewers share similar critiques of your work, they have also identified several novel and significant issues that need to be addressed. 

We look forward to receiving your revised manuscript.

Kind regards,

Brittany N. Florkiewicz, Ph.D.

Academic Editor

PLOS ONE

3. Please upload a copy of Supporting Information Figure/Table/etc. Figure S1 which you refer to in your text on page 19.

4. Please include captions for your Supporting Information files at the end of your manuscript, and update any in-text citations to match accordingly. Please see our Supporting Information guidelines for more information: http://journals.plos.org/plosone/s/supporting-information .

Reviewers' comments:

Reviewer's Responses to Questions

**Comments to the Author**

1. Is the manuscript technically sound, and do the data support the conclusions?

Reviewer #1: Yes

Reviewer #2: Partly

Reviewer #3: Partly

2. Has the statistical analysis been performed appropriately and rigorously?

Reviewer #1: Yes

Reviewer #2: Yes

Reviewer #3: Yes

3. Have the authors made all data underlying the findings in their manuscript fully available?

Reviewer #1: Yes

Reviewer #2: Yes

Reviewer #3: Yes

4. Is the manuscript presented in an intelligible fashion and written in standard English?

Reviewer #1: Yes

Reviewer #2: Yes

Reviewer #3: Yes

Reviewer #1: Manuscript Number: PONE-D-25-25871

The authors present an important and timely study on body awareness in dogs. While previous research has attempted to explore this aspect of canine cognition, the authors provide clearer evidence by employing a two-choice paradigm and testing dogs with varying levels of training experience. This study is likely to make a meaningful contribution to the field of animal cognition, particularly in understanding self-awareness. However, I have several questions and suggestions that I believe could help strengthen and clarify the manuscript.

Major Comments

The title should be reconsidered to avoid potential misunderstanding. While “body-size perception” is an appropriate and well-established term, the use of “body-shape perception” and “body-solidity perception” may be problematic. To the best of my knowledge, the term “body-shape perception” is not commonly used in animal cognition research, including previous studies on body awareness. Moreover, the distinction between body size and body shape appears ambiguous and potentially redundant, as the two concepts may substantially overlap in meaning. Furthermore, “body-solidity perception” is inaccurate, as the subjects were perceiving the solidity of the covering materials, not their own bodies.

In the analysis, all combinations of windows were treated equally; however, some should be interpreted separately. There are two types of combinations: one where both doors are available (e.g., the Horizontal large enough and the Vertical medium size), and another where one side is unavailable (e.g., the Horizontal large enough and the Vertical too small). While it is reasonable to define the larger and the particular shape as the correct choice, it is problematic to analyze both types of conditions in the same way. In the former case, the “incorrect” option is still “available,” whereas in the latter case, the “incorrect” option is actually “unavailable.” This difference affects how the dogs’ choices should be interpreted.

In the Results section, it would be helpful to include a comparison between the two-door choices as well as between the trained and untrained groups. While p-values of the former comparisons are reported, those for the latter are not. A comparison between the training conditions is essential for clarifying how prior experience contributes to the development of body awareness.

Reporting the exact choice ratios would enhance the interpretability of the results, rather than relying solely on p-values. I also recommend presenting some of the findings as figures to improve clarity, as too many tables can make the data difficult to follow. Additionally, the current table format does not clearly indicate which door was chosen in each trial, making it difficult for readers to quickly grasp the key outcomes.

To support the claim that animals possess self-awareness, some evidence of self-directed behavior is generally required (Gallup & Anderson 2018, Behavioural Processes; Gallup & Anderson 2020, Psychology of Consciousness: Theory, Research, and Practice.; Kakrada & Colombo 2022, Learning and Motivation). Self-directed behavior refers to actions in which animals inspect or interact with their own body, such as mirror-guided self-exploration (Gallup, 1977, American Psychologist). If any potentially relevant behaviors were observed—such as hesitation or re-examination of body size before attempting to pass through a narrow opening—I suggest mentioning them, even if only as qualitative observations.

Miner Comments

Title

Line 1: As noted in the Major Comments, “body-shape perception” should be incorporated into “body-size perception,” as the distinction between the two is unclear. Additionally, “body-solidity perception” is an inaccurate term, as it refers to the perception of the covering materials rather than the subjects’ own body.

Line 2: The term “pet” is likely unnecessary, as the majority of studies involving dogs already use pet dogs as subjects. Unless there is a specific reason to distinguish them from working or shelter dogs, the term can be omitted to improve conciseness.

Abstract

Line 28: It is unclear what exactly became “faster” or “slower.” Specifying the exact parameter being described would improve the clarity of the results.

Introduction

Line 70: This paragraph and the one that follows could be condensed and integrated to reduce redundancy.

Line 81: The citation style appears to be inconsistent with the journal’s guidelines.

Line 83: A citation is needed to support this statement.

Line 108: If the final “building block” has been demonstrated, is it still necessary to empirically test the two preceding ones?

Ethical statement

Line 167: How much time did each subject spend participating in the experiment?

Materials and methods

Line 231: What was the inter-trial interval used in the experiment?

Line 233: How long was the “short break” mentioned in the procedure?

Line 239: Does this definition of “first approach” follow any precedent in previous research? If this criterion was developed specifically for this study, it would be important to provide supporting evidence or justification to ensure its validity.

Line 242: Why was latency not measured from the moment the owner unleashed the dog? Measuring latency from that point could provide insight into whether the subject approached the door immediately or hesitated before making a decision—an important distinction that can help detect behaviors that may indicate self-directed processing.

Line 253: Why was the rationale for including the z-transformed trial number as a variable in Experiment 2?

Experiment 1: Table 1: In the “Vertical” column, the label “C” should be corrected to “CW” for accuracy.

Experiment 1: Results

Line 330: Does the term “training” refer to agility training specifically, or to the training phase within the experimental procedure?

Line 359: This combination should not be interpreted in the same way as the others, as the subjects can available both windows in this case, unlike in the other two combinations.

Line 363: The interpretive value of this measurement may be limited, as the subjects are capable of passing through both “Large” and “Medium” windows. If both options are equally accessible, it becomes difficult to determine whether the subjects’ choices reflect body-size awareness or are simply arbitrary.

Experiment 1: Discussion

Paragraph 3, line 5: To more clearly assess this possibility, a combination of the Circular window with either the Horizontal or Vertical too small should be tested. These options appeared to be less preferred by the subjects.

Experiment 2: Object solidity

Subjects: How long was the interval between Experiment 1 and Experiment 2 for subjects that participated in both experiments?

Experiment 2: Results

Paragraph 2, line 10: Should the final combination be 1 and 3 rather than 1 and 2?

Experiment 2: Discussion

Paragraph 1, line 5: Are the relevant statistical results reported in the Results section? If not, please consider including them to support the interpretation of the findings and ensure consistency with the hypotheses or claims made in the Discussion.

Paragraph 2, line 3: Does agility training include any components that involve learning about material solidity?

Paragraph 2, line 10: Providing details of a representative agility training program would help readers better understand and contextualize the results. I strongly recommend including this information either in the main manuscript or as supplementary material.

General Discussion

Paragraph 1, line 1: In the case of dogs, the olfactory self-recognition test has been developed as a means to investigate self-awareness (Horowitz 2017, Behavioural Processes). This study may be worth citing, as it provides an alternative approach to assessing self-awareness in dogs. Together with the accompanying comment article, it offers valuable insights into how self-awareness in animals can be examined (Gallup & Anderson 2018, Behavioural Processes).

Paragraph 3, line 12: It would be helpful to clarify the relationship between the “first approach” and the “first attempt.” What distinguishes trials in which subjects attempted to go through the same window they first approached from those which they did not? How many times do subjects approach the windows before making their “first attempt”? Are all subjects “walked through” the window on their “first attempt”?

Paragraph 4, line 8: The logical link between this idea and the preceding statement is unclear. Please consider providing additional explanation to clarify the relationship.

Table S1: The raw p-values should be omitted; only the adjusted p-values should be reported.

References: The published month and date may be unnecessary. The title should follow sentence case, capitalizing only the first word and any proper nouns. Scientific names should be italicized.

Reviewer #2: In this study, two groups of dogs (untrained pet dogs vs. agility-trained dogs) were compared in a two-choice task requiring them to pass through one of two openings (Experiment 1: variation in size/shape; Experiment 2: variation in solidity). The authors conclude that: (i) dogs correctly assess their own body size; (ii) evidence regarding shape perception is less conclusive; (iii) dogs partially discriminate the solidity of obstacles; and (iv) experience (i.e., agility training) primarily affects decision speed and, to a lesser extent, shape perception.

This study offers a useful extension of the work by Lenkei et al. (2021), by testing multiple “building blocks” of the ecological self within the same species, and by introducing a two-option paradigm that allows for better control over motivational factors. The overall contribution is promising, but several major issues need clarification or improvement before publication.

- The categorization of dogs as “agility-trained” is based on relatively broad criteria (regular training and participation in A1-level competitions), which likely includes individuals with highly variable levels of intensity, frequency, and duration of training. To better characterize this group and assess its internal heterogeneity, it would be helpful to include additional information such as average weekly training time, age at training onset, and date of most recent competition participation.

- The authors conclude that agility-trained dogs exhibit enhanced shape perception. However, this conclusion appears to overreach the empirical findings. The supporting results are limited to a single statistically significant condition, with a small effect size, and do not justify a clear claim of increased perceptual accuracy. The data more plausibly reflect faster decision-making or motor familiarity in specific configurations, rather than a finer or more conscious representation of body shape. It would be advisable to temper this conclusion in the discussion.

- The analyses rely on an a priori categorization of choices as “correct” or “incorrect,” based on what the authors define as the optimal solution for the animal (e.g., a larger opening or a softer material). This approach carries the risk of imposing an anthropocentric decision framework, assuming that the dog’s perception of its own body necessarily aligns with human preferences or strategies in similar contexts. However, dogs’ individual preferences, contextual motivations, or sensory interpretations may diverge from this framework. It would be useful for the authors to acknowledge this limitation and to consider, in the discussion, the possibility that some choices judged as “suboptimal” from a human perspective may nonetheless reflect valid body representation from the animal’s point of view. A complementary approach could involve coding and analyzing bodily adjustments or hesitation behaviors as direct indicators of body perception, independently of the final choice.

- Given that one of the study’s stated aims is to evaluate the dog as a model for the systematic study of body and self-awareness (i.e., “ecological self”), the discussion would benefit from a more explicit comparative perspective involving other species. Specifically, previous studies using similar paradigms in elephants, rats, or birds should be mobilized not only qualitatively but also quantitatively, by citing observed effect sizes when available. This would help situate canine performance within a more rigorous comparative framework and strengthen the argument for their suitability as a model species. Such inclusion would also help clarify the potential species-specific contributions of ecological niche or life experience to the observed patterns.

Reviewer #3: The manuscript presents novel data on the phenomenon of body-awareness in dogs. Although dogs have previously been the subject of similar research on multiple occasions, the authors' results provide valuable insights for advancing a comprehensive understanding of self-awareness across animal species.

Nevertheless, I have several critiques regarding the "Introduction," "General Discussion," and the validity justification of the experimental methodology.

Point 1. Lines 79–92: While the authors provide a general overview of methodology for studying body size awareness across animal species, the literature review is notably limited. The phenomenon of body size awareness—tested using body-as-obstacle tasks—has been documented in the following species (beyond those cited):

Budgerigars

Schiffner, I.; Vo, H.D.; Bhagavatula, P.S.; Srinivasan, M.V. Minding the gap: In-flight body awareness in birds. Front. Zool. 2014, 11, 64

Hooded crows

Khvatov, I.A.; Smirnova, A.A.; Samuleeva, M.V.; Ershov, E.V.; Buinitskaya, S.D.; Kharitonov, A.N. Hooded Crows (Corvus cornix) May Be Aware of Their Own Body Size. Front. Psychol. 2021, 12, 769397.

Ferrets

Khvatov, I.A.; Sokolov, A.Y.; Kharitonov, A.N. Ferrets (Mustela furo) Are Aware of Their Dimensions. Animals 2023, 13, 444.

Domestic cats

Pongrácz, P. Cats are (almost) liquid!—Cats selectively rely on body size awareness when negotiating short openings. iScience 2024, 27, 110799.

Wistar rats

Khvatov, I.A.; Ganza, P.N.; Kharitonov, A.N.; Samuleeva, M.V. Wistar Male Rats (Rattus norvegicus domestica) Are Aware of Their Dimensions. Animals 2024, 14, 3384.

Hermit crabs

Sonoda, K.; Moriyama, T.; Asakura, A.; Furuyama, N.; Gunji, Y.P. Can Hermit Crabs Perceive Affordance for Aperture Crossing? In Proceedings of the European Conference on Complex Systems 2012, Brussels, Belgium, 3–7 September 2012; Springer: Cham, Switzerland, 2013; pp. 553–557.

Krieger, J.; Hörnig, M.K.; Laidre, M.E. Shells as ‘extended architecture’: To escape isolation, social hermit crabs choose shells with the right external architecture. Anim. Cogn. 2020, 23, 1177–1187.

Recommendation for authors.

Integrate references to Schiffner et al. (2014), Khvatov et al. (2021, 2023, 2024), Pongrácz (2024), and Krieger et al. (2020). Highlight methodological parallels (e.g., aperture-crossing paradigms) to position this study within a broader comparative framework and enrich interpretation of your findings.

Point 2. Lines 94–96. The authors claim that "the only study" investigating animals’ ability to relate bodily properties to environmental features was conducted on rats (assessing weight sensitivity). I disagree with this assertion.

While the cited rat study (14) explicitly demonstrated probing behavior to test substrate solidity, the capacity to integrate body properties with object features is inherently tested in any aperture-crossing paradigm—whether animals: decide to pass through a single opening, or select a viable opening among multiple options.

For example:

Birds folding wings mid-flight when traversing narrow vertical slits demonstrate pre-contact anticipation of collisions, directly relating body dimensions to aperture parameters before physical interaction (Schiffner et al., 2014).

Rats or ferrets immediately approaching and traversing a size-appropriate opening (when alternatives are impassable) similarly reveal this cognitive integration (Khvatov et al., 2023, 2024).

Recommendation for authors.

Revise to: "Few studies explicitly test animals’ ability to integrate bodily properties with object solidity. While rats (14) demonstrated substrate probing, aperture-crossing paradigms in birds and mammals inherently require this integration." Supporting citations: Schiffner et al. (2014), Khvatov et al. (2023, 2024), Pongrácz (2024).

Point 3. Lines 141–145. The authors argue that a two-opening paradigm is superior to a single-opening design. However, a critical question remains unaddressed: Why was a three-opening setup not employed? This alternative is methodologically justified for two primary reasons:

Ecological validity vs. cognitive load. Experiment 1 tested three aperture sizes; Experiment 2 used three barrier materials. A three-opening design would enable simultaneous presentation of all variants, granting subjects fuller choice—enhancing ecological validity. Counterpoint: three openings may overcomplicate the task, demanding higher attentional focus and potentially confounding results.

Methodological precedent. Three-opening paradigms have been successfully implemented in analogous studies (e.g., ferrets: Khvatov et al., 2023; rats: Khvatov et al., 2024), testing aperture shapes/sizes comparable to this work.

While I do not assert that three openings are inherently superior, the authors should justify their design choice relative to established methods. Results should also be contextualized against prior findings using multi-opening paradigms.

Recommendation for authors.

Explicitly state the rationale for selecting two openings. For example: "We retained the two-choice paradigm to: (a) Maintain comparability with prior canine body-awareness research (Lenkei et al., 2020); (b) Mitigate cognitive overload during extended 18-trial sessions. We acknowledge that while a three-opening design could enable direct comparison of all conditions, it risks reducing decision clarity. Future studies could test this adaptation."

Point 4. Interpretation of Experiment 1 Results. Page 12. The authors claim: "Animals did correctly assess their size, as they approached, attempted, and went through the correct barrier above chance". This conclusion lacks sufficient empirical support. My rationale:

1. Behavioral sequence matters. Under high motivation, first-approach behavior is a stronger indicator of body size awareness than first-attempt or crossing choices. Animals may approach one aperture, investigate it, then cross through another (e.g., approach oversized → cross medium-sized). Critical metric: frequency of trials where the first-approached aperture was immediately crossed without prior investigation of alternatives. This is unreported.

2. Ambiguous first-approach data. Horizontal size effect: "Both groups approached the bigger option more often" (line 357) → Confirms preference for larger apertures, not selection of size-appropriate ones. Vertical size effect: Preference for large vs. small apertures only → No evidence dogs distinguished large vs. medium or medium vs. small.

3. Contrast with robust paradigms. Studies in ferrets/rats (Khvatov et al., 2023, 2024) used three-aperture designs: two impassably large openings and one passable small opening. Subjects preferentially approached and crossed passable apertures, proving size-appropriate selection—not just "bigger is better." Current data cannot confirm dogs chose passable over merely larger openings.

Recommendation for authors.

Reanalyze data to track:

1. First-approach-to-immediate-crossing sequences: How often did dogs cross the first aperture they approached?

2. Individual-level cases: Document trials where dogs approached oversized apertures but crossed medium-sized ones.

Point 5. Demographic Variables in Statistical Models. The manuscript reports subjects' sex and age (with notably high age variance). However, the statistical analysis omits these variables as covariates in the models—a significant oversight requiring correction.

Recommendation for authors.

Refit all GLMM and COXME models including age (linear or log-transformed) and sex as covariates. Report whether the agility training effect persists after controlling for these demographic factors.

Point 6. Terminology: "Correct Choice" in Statistical Analysis (lines 307–312). The authors define a "correct choice" for each aperture combination in Experiment 1. This framing is methodologically unsound, particularly given that results contradicted initial hypotheses. Crucially:

1. Functional success ≠ predicted preference: A "correct" choice should reflect task success (reward attainment), not alignment with researcher predictions.

2. Example: Preferring a horizontal over vertical aperture is not "incorrect" if both yield rewards—it simply reflects subject preference.

3. Parallel issue in Experiment 2: The same problematic framing applies to barrier solidity choices.

Recommendation for authors.

Replace "correct choice" with "predicted choice" throughout Methods and Results. Explicitly state: "In combinations where both options were passable (e.g., horizontal vs. vertical), 'non-predicted' choices still enabled reward acquisition."

Point 7. Fatigue and Motivation Control in Experiment 2. Dogs encountered barriers requiring physical effort (tearing paper, pushing aside foliage) or presenting impassable obstacles (wood). The session structure—18 test trials plus 6 familiarization trials—imposes significant cognitive and physical load. This raises concerns about: progressive motivational decline; fatigue-induced performance decay (especially for latency and success probability within time limits). While breaks were mentioned, their effectiveness was not empirically verified.

Recommendation for authors.

1. Analyze trial-order effects: Segment sessions into thirds (early/mid/late) and compare key metrics (latency, success rate).

2. Include trial number as a covariate: Apply to all GLMM/COXME models in both experiments (currently partial for Exp2 only).

3. Acknowledge limitation: Discuss fatigue as a potential confounding factor in the "Limitations" section.

Point 8. Interpretation of Circular Aperture Avoidance (Experiment 1). The authors hypothesize that avoidance of circular openings stems from "the necessity to jump with hind legs and the risk of hitting the brim." However, this interpretation lacks direct empirical support.

Recommendation for authors:

1. Reframe as a hypothesis: Explicitly state this explanation as one possible interpretation rather than a conclusion.

2. Conduct supplemental video analysis (if feasible): Quantify hind-leg jumping behavior during circular aperture crossings; Compare collision frequency (brim contact) between circular vs. other aperture shapes.

3. Acknowledge limitations: If video data is insufficient, explicitly note in "Limitations" that "the cause of circular aperture avoidance remains an unverified hypothesis."

Point 9. Concern: Definition of "Medium" Apertures in Experiment 1. The authors define medium openings as "exactly the size the animal needed to walk through" (lines 272–273). However, this rigid definition overlooks critical biological and behavioral variability:

1. Individual differences in fur density/volume;

2. Posture-dependent body compression/expansion during passage;

3. Risk: Some "medium" apertures may have been functionally impassable or uncomfortable for certain dogs/postures, potentially explaining: absence of large vs. medium preference; observed aperture collisions.

Recommendation for authors:

1. Specify sizing protocol: Detail how "medium" was calculated (e.g., body measurement + X% clearance? Exact withers height?).

2. Discuss limitations: Address whether sizing accommodated: fur/coat volume; dynamic posture adjustments; inter-individual variability.

3. Acknowledge as potential confound: Cite this as a plausible explanation for null results in size comparisons.

Point 10. Comparison with Lenkei et al. (2021) on Aperture Shape. Lenkei et al. (the primary cited study) concluded that dogs (especially short-legged breeds) do not rely on shape perception, successfully traversing both horizontal and vertical slits through postural adaptation. By contrast, this study:

1. Expected preference for circular/vertical openings over horizontal ones;

2. Found no strong shape preference (beyond circular avoidance), except a trend toward horizontal openings in agility-trained dogs

Recommendation for authors.

Expand the General Discussion to reconcile these findings:

1. Emphasize that both studies demonstrate dogs' flexibility in navigating diverse shapes, but this work used a choice paradigm (vs. single-aperture trials in Lenkei et al.).

2. Note that the absence of shape preference (excluding circular) aligns with Lenkei et al.'s thesis of body-awareness plasticity.

3. Highlight the intriguing trend among agility dogs toward horizontal openings as task-specific behavior—potentially stemming from frequent tunnel/under-barrier training. Position this as a key question for future research.

Point 11. Agility Training Effect in Experiment 2. The finding that untrained dogs outperformed agility-trained dogs in solidity-based choices is unexpected. While authors propose hypotheses (e.g., trained dogs' broader material experience reducing perceived solidity differences; social inhibition; conditioned avoidance of barrier destruction), this result demands cautious interpretation.

Recommendation for authors:

1. Highlight the counterintuitive nature: Emphasize this as an intriguing finding requiring cautious interpretation.

2. Propose targeted follow-ups: test dogs without owners present to evaluate "social inhibition" (though logistically complex); use less destructible barriers (e.g., push-flaps vs. tearable paper) to assess "avoidance of destruction" hypothesis.

3. Introduce attentional focus hypothesis: Suggest that agility training may prioritize processing trajectory/speed cues over detailed solidity assessment of static barriers—contrasting with untrained dogs' more exploratory approach.

Minor Recommendations

Point 12. Keywords: Add "body size awareness" to the list. This term accurately reflects a core focus of the study and enhances discoverability.

Point 13. Line 81: Standardize citation formatting to match the journal’s style guide. The current inconsistency deviates from the full citations used elsewhere.

**Do you want your identity to be public for this peer review?** For information about this choice, including consent withdrawal, please see our Privacy Policy

Reviewer #1: No

Reviewer #2: **Yes: ** Melina Cordeau

Reviewer #3: **Yes: ** Ivan Khvatov

---

## [Author Response · Author response to Decision Letter 1]

3 Sep 2025

Reviewer #1: Manuscript Number: PONE-D-25-25871

The authors present an important and timely study on body awareness in dogs. While previous research has attempted to explore this aspect of canine cognition, the authors provide clearer evidence by employing a two-choice paradigm and testing dogs with varying levels of training experience. This study is likely to make a meaningful contribution to the field of animal cognition, particularly in understanding self-awareness. However, I have several questions and suggestions that I believe could help strengthen and clarify the manuscript.

Major Comments

The title should be reconsidered to avoid potential misunderstanding. While “body-size perception” is an appropriate and well-established term, the use of “body-shape perception” and “body-solidity perception” may be problematic. To the best of my knowledge, the term “body-shape perception” is not commonly used in animal cognition research, including previous studies on body awareness. Moreover, the distinction between body size and body shape appears ambiguous and potentially redundant, as the two concepts may substantially overlap in meaning. Furthermore, “body-solidity perception” is inaccurate, as the subjects were perceiving the solidity of the covering materials, not their own bodies.

Our response: the title has been changed to: Influence of agility training on body-size and object solidity perception in pet dogs

In the analysis, all combinations of windows were treated equally; however, some should be interpreted separately. There are two types of combinations: one where both doors are available (e.g., the Horizontal large enough and the Vertical medium size), and another where one side is unavailable (e.g., the Horizontal large enough and the Vertical too small). While it is reasonable to define the larger and the particular shape as the correct choice, it is problematic to analyze both types of conditions in the same way. In the former case, the “incorrect” option is still “available,” whereas in the latter case, the “incorrect” option is actually “unavailable.” This difference affects how the dogs’ choices should be interpreted.

Our response: In our analysis, we tested if the animals had a preference for one side or the other, i.e. do animals pick one opening significantly above chance when two openings were presented. Importantly we looked at the first side approached and first side attempted – so before they actually experienced that one choice was not really available after all (because they did not fit through it). To be able to do so, we defined one opening as “correct” (coded as 1) and the other as “incorrect” (coded as 0), which matches the predictions of our hypotheses. In each model, we included the effect of the “combination number” (i.e. the actual combination presented), which tells us if there is a difference between each combination. Therefore, we did test for this specific difference between the two types of combinations presented when analysing which opening they first approached and attempted. For the actual opening they walked through, we only analysed the combinations where both options were passable.

In the Results section, it would be helpful to include a comparison between the two-door choices as well as between the trained and untrained groups. While p-values of the former comparisons are reported, those for the latter are not. A comparison between the training conditions is essential for clarifying how prior experience contributes to the development of body awareness.

Our response: We added the p-values in text where appropriate (in comparisons 3 and 4 of Experiment 1).

Reporting the exact choice ratios would enhance the interpretability of the results, rather than relying solely on p-values. I also recommend presenting some of the findings as figures to improve clarity, as too many tables can make the data difficult to follow. Additionally, the current table format does not clearly indicate which door was chosen in each trial, making it difficult for readers to quickly grasp the key outcomes.

Our response: We provided additional graphs in the result section to improve clarity. The graphs show the ratio of choices per training group per condition. The ratios were already reported as percentages of trials where they chose the predicted correct option in the tables along with the respective confidence intervals. The predicted correct choices are explained in the statistical analysis section and at the end of the introduction.

To support the claim that animals possess self-awareness, some evidence of self-directed behavior is generally required (Gallup & Anderson 2018, Behavioural Processes; Gallup & Anderson 2020, Psychology of Consciousness: Theory, Research, and Practice.; Kakrada & Colombo 2022, Learning and Motivation). Self-directed behavior refers to actions in which animals inspect or interact with their own body, such as mirror-guided self-exploration (Gallup, 1977, American Psychologist). If any potentially relevant behaviors were observed—such as hesitation or re-examination of body size before attempting to pass through a narrow opening—I suggest mentioning them, even if only as qualitative observations.

Our response: We did not observe any self-directed behaviours that would be worth mentioning. However, as mentioned in the introduction, we focused on a very small aspect of self-awareness, which should be achievable by any animal with active locomotion. This aspect of self-awareness probably does not require self-directed behaviours and thus could be achieved simply from sensory information gathered through interaction with the environment. We did not want to claim with this study that animals possess self-awareness but rather focus on perception of physical attributes.

Miner Comments

Title

Line 1: As noted in the Major Comments, “body-shape perception” should be incorporated into “body-size perception,” as the distinction between the two is unclear. Additionally, “body-solidity perception” is an inaccurate term, as it refers to the perception of the covering materials rather than the subjects’ own body.

Our response: We changed the title accordingly. We also replaced any mention of body-solidity perception with object solidity perception. We would still prefer to keep the term shape perception. Although shape is technically incorporated in size perception, as shape is just size in different directions, it was not explicitly tested previously if animals perceive these measurements differently. This distinction between size and shape also helps separate different conditions.

Line 2: The term “pet” is likely unnecessary, as the majority of studies involving dogs already use pet dogs as subjects. Unless there is a specific reason to distinguish them from working or shelter dogs, the term can be omitted to improve conciseness.

Our response: Our lab often works with pet dogs, free-ranging dogs, and dogs raised in a wild park (pack dogs). Therefore, specifying that these are pet dogs may be important to some readers.

Abstract

Line 28: It is unclear what exactly became “faster” or “slower.” Specifying the exact parameter being described would improve the clarity of the results.

Our response: rephrased this line

Introduction

Line 70: This paragraph and the one that follows could be condensed and integrated to reduce redundancy.

Our response: the condensed paragraph now reads as follows:

To our knowledge only a few studies investigated the ability to put the properties of one’s body in relation with the properties of other objects in their environment [1–9]. Awareness of one's own body size is the most commonly studied property of body as an object, and it is usually studied in the context of species whose males fight for resources (territory, potential mates, etc.) (e.g. [10,11]). In that context, it is beneficial for an individual to have a perception of one’s own relative size to be able to assess how much bigger or smaller the opponent is. Only few studies so far tested body size perception in the context of foraging and moving through the environment [1–9]. The paradigm most frequently used is presenting the animals with gaps of different sizes to go through or walk over. If the animals are aware of their body size, they should choose to walk through bigger gaps when available, adjust their bodies accordingly as the gaps get smaller, and be more hesitant to attempt going through small gaps. Alternatively, some studies used a single gap paradigm in which case the latencies to approach or reach the reward and the behavioural modifications (e.g. body pose, ducking the head, etc.) were taken as the measure of the awareness of body size. Animals across taxa seem to be aware of their body size and adjust their behaviour as they move through or walk over differently sized gaps [1–4,12], suggesting that understanding of one’s own size is indeed crucial for successful navigation through the environment. The only study that focused on other properties of body as an object (other than shape and size) was carried out on brown rats [5]. The rats had to cross one of the three bridges presented in the enclosure to reach a reward, however only one bridge was fixated, while the other two were loose (acting like a see-saw). The brown rats seem to be able to put their own body weight in the relation to the solidity of the object they walk on. However, out of 25 animals, only 8 developed a probing behaviour (tapping the bridge with their paws) to discriminate between bridges, 12 gave up after a few falls, and the remaining 5 did not develop the correct strategy but kept on trying. This raises an interesting point that although the rats may be aware of their weight and how it interacts with a loose bridge, it does not mean that animals develop the correct strategy to test the properties of another object in the environment and, as mentioned by the authors, some rats might have perceived the falling off the bridge as too big of a risk to invest more energy to finding a solution.

Line 81: The citation style appears to be inconsistent with the journal’s guidelines.

Our response: corrected

Line 83: A citation is needed to support this statement.

Our response: the phrasing here was changed as we condensed the two paragraphs as mentioned in the point above. The needed references were added where appropriate. See the paragraph added in the response above.

Line 108: If the final “building block” has been demonstrated, is it still necessary to empirically test the two preceding ones?

Our response: Building blocks may differ between species as they may use different senses to experience them, so within species it would be important to define each of the building blocks to be able to explain how they develop. Therefore, it is hard to define how building blocks may relate to one another without testing for each of them separately. This is also in line with our results that shape and size may be assessed separately.

Ethical statement

Line 167: How much time did each subject spend participating in the experiment?

Our response: specified approximate duration of the experiment of 40 minutes (line 208*)

*the lines specified here and throughout this document refer to revised manuscript with tracked changes

Materials and methods

Line 231: What was the inter-trial interval used in the experiment?

Our response: The inter-trial interval was the time needed for the experimenter to setup the next trial and answer any possible questions that the owner may ask after the trial (often asking if they are doing everything as instructed and how their dog is doing in the study). This usually took a few minutes (1-5 minutes). The next trial started as soon as possible.

Line 233: How long was the “short break” mentioned in the procedure?

Our response: duration specified in line 274

Line 239: Does this definition of “first approach” follow any precedent in previous research? If this criterion was developed specifically for this study, it would be important to provide supporting evidence or justification to ensure its validity.

Our response: in the study by Lenkei et al. (2019) and by Horrowitz at al. (2021) they also defined approach (but did not provide exact measurement of distance from the opening). In the studies by Khvatov et al, they defined the approach as the situation where the tip of the nose/beak of the animal is within 10 cms from the opening. We used a well-established measurement (body length) used in dog studies as a consistent measurement for each subject as different subjects may significantly vary in size(to name a few studies using this measurement [1–5]).

Line 242: Why was latency not measured from the moment the owner unleashed the dog? Measuring latency from that point could provide insight into whether the subject approached the door immediately or hesitated before making a decision—an important distinction that can help detect behaviors that may indicate self-directed processing.

Our response: We added the analysis of latency to approach in the results section of both experiments as suggested (before latency to reach reward). We added the required definition in statistical analysis section(s).

Line 253: Why was the rationale for including the z-transformed trial number as a variable in Experiment 2?

Our response: In experiment 2 there were only 3 combinations and they were presented with 6 trials per combination. The effect of the trial number here provides us with the information about the performance over time. In Experiment 1 each trial was a unique combination randomized across session, meaning the trial number does not provide additional information. We added the rationale for z-transforming the trial number in the statistical analysis section (line 295).

Experiment 1: Table 1: In the “Vertical” column, the label “C” should be corrected to “CW” for accuracy.

Our response: corrected

Experiment 1: Results

Line 330: Does the term “training” refer to agility training specifically, or to the training phase within the experimental procedure?

Our response: the term “training” refers to agility training. The same term is mentioned in the statistical analysis (line 292). We added a clarification in the brackets that training as a variable refers to the two groups of dogs (agility trained or untrained) (line 292).

Line 359: This combination should not be interpreted in the same way as the others, as the subjects can available both windows in this case, unlike in the other two combinations.

Line 363: The interpretive value of this measurement may be limited, as the subjects are capable of passing through both “Large” and “Medium” windows. If both options are equally accessible, it becomes difficult to determine whether the subjects’ choices reflect body-size awareness or are simply arbitrary.

Our response (including previous point): These points are covered by the second major point. Copy of our response there: In our analysis, we tested if the animals had a preference for one side or the other, i.e. do animals pick one opening significantly above chance when two openings were presented. Importantly we looked at the first side approached and first side attempted – so before they actually experienced that one choice was not really available after all (because they did not fit through it). To be able to do so, we defined one opening as “correct” (coded as 1) and the other as “incorrect” (coded as 0), which matches the predictions of our hypotheses. In each model, we included the effect of the “combination number” (i.e. the actual combination presented), which tells us if there is a difference between each combination. Therefore, we did test for this specific difference between the two types of combinations presented when analysing which opening they first approached and attempted. For the actual opening they walked through, we only analysed the combinations where both options were passable.

Experiment 1: Discussion

Paragraph 3, line 5: To more clearly assess this possibility, a combination of the Circular window with either the Horizontal or Vertical too small should be tested. These options appeared t

---

## [Decision Letter · Decision Letter 1]

15 Sep 2025

Dear Dr. Staric,

Thank you for submitting your manuscript to PLOS ONE and for your diligent work on the manuscript revisions. Reviewers #1 and #2 have expressed their satisfaction with these changes. However, Reviewer #3 has raised some important concerns regarding your data analysis and data collection protocols. These points can be addressed fairly quickly, so I will be recommending a minor revision for the manuscript.

We look forward to receiving your revised manuscript.

Kind regards,

Brittany N. Florkiewicz, Ph.D.

Academic Editor

PLOS ONE

Journal Requirements:

Reviewers' comments:

Reviewer's Responses to Questions

**Comments to the Author**

Reviewer #1: All comments have been addressed

Reviewer #2: All comments have been addressed

Reviewer #3: All comments have been addressed

2. Is the manuscript technically sound, and do the data support the conclusions?

Reviewer #1: Yes

Reviewer #2: Yes

Reviewer #3: Partly

3. Has the statistical analysis been performed appropriately and rigorously?

Reviewer #1: Yes

Reviewer #2: (No Response)

Reviewer #3: No

4. Have the authors made all data underlying the findings in their manuscript fully available?

Reviewer #1: Yes

Reviewer #2: (No Response)

Reviewer #3: Yes

5. Is the manuscript presented in an intelligible fashion and written in standard English?

Reviewer #1: No

Reviewer #2: (No Response)

Reviewer #3: Yes

Reviewer #1: (No Response)

Reviewer #2: (No Response)

Reviewer #3: Thank you for addressing most of my comments and for the corresponding revisions. A few points, however, remain insufficiently addressed. Below I clarify the rationale and offer specific suggestions for revision.

Point 5 — Demographic covariates

Thank you for the clarification. Here is why we ask you to account for sex and age in the models, even if these variables were not part of the a priori hypothesis.

• “We had no hypotheses about sex/age.” Adding covariates is not about testing a new hypothesis; it is a standard way to control potential confounding and to improve the precision of the main treatment estimate. Modern modeling guidance emphasizes that adjustment for relevant predictors reduces noise and yields more efficient standard errors (SEs) and confidence intervals (CIs). For age, please model the effect flexibly (e.g., restricted cubic splines) rather than as a strictly linear term.

See: Frank E. Harrell Jr., Regression Modeling Strategies.

• “Groups are similar.” Similar means at the group level do not preclude individual-level confounding, particularly when age variance is high; adjusting for informative covariates increases efficiency of the treatment effect even under randomization/balanced groups.

• “All dogs ≥14 months; body image is consolidated.” Physical maturity does not eliminate between-individual differences in attention, learning speed, and cognitive flexibility in adult dogs; age-related variation in these domains is well documented.

See: Wallis LJ, Range F, Müller CA, Serisier S, Huber L, Virányi Z (2014). Lifespan development of attentiveness in domestic dogs: drawing parallels with humans. Frontiers in Psychology, 5:71. https://doi.org/10.3389/fpsyg.2014.00071

• Empirical evidence for sex effects in canine cognition. Several studies report sex differences in dogs’ cognitive/spatial tasks.

See:

• Fugazza C, Mongillo P, Marinelli L. Sex differences in dogs’ social learning of spatial information. Animal Cognition (2017) 20(4):789–794.

• Mongillo P, Scandurra A, D’Aniello B, Marinelli L. Effect of sex and gonadectomy on dogs’ spatial performance. Applied Animal Behaviour Science (2017) 191:84–89.

• Junttila S, Huohvanainen S, Tiira K. Effect of sex and reproductive status on inhibitory control and social cognition in the domestic dog (Canis familiaris). Animals (2021) 11:2448.

• Reporting standards. ARRIVE 2.0 Explanation & Elaboration (Item 7, Statistical methods) explicitly states that nuisance variables should be considered in the analysis (e.g., as covariates); sex and age are listed among factors influencing outcomes. In addition, the NIH SABV policy expects sex to be factored into design, analysis, and reporting in vertebrate animal studies. Please either provide analyses with these covariates or offer a strong justification for excluding them.

See: Percie du Sert N. et al. (2020). Reporting animal research: Explanation and elaboration for the ARRIVE guidelines 2.0. PLOS Biology 18(7): e3000411.

Requested action:

— Refit all generalized linear mixed-effects models (GLMMs) and mixed-effects Cox models (CoxME) including sex and age as fixed covariates; model age with restricted cubic splines (3–5 knots). Report whether the agility-training effect persists and how SEs/CIs change.

— Provide a sensitivity analysis (models with vs. without these covariates); if feasible, exploratory training×age and training×sex interactions with appropriate caveats.

— Reflect these choices transparently in Statistical methods in line with ARRIVE 2.0 (E&E, Item 7).

Point 7 — Fatigue/motivation and trial-order effects

Thank you for the clarification. A subject random intercept adjusts baseline between-dog differences but does not model within-session trends (learning, fatigue, motivational decline). Such trends should be accounted for by an explicit trial index predictor; for inference, please treat trial as a continuous variable with a flexible functional form (e.g., restricted cubic splines) and, where possible, include a by-dog random slope for trial. Person-mean centering of the trial index (within-subject centering) is recommended to cleanly separate within- and between-dog components. This approach is standard for repeated-measures designs and avoids anti-conservative inference; splitting into thirds is unnecessary for the main model and can be reserved for descriptive checks. For time-to-success outcomes, include trial in Cox/CoxME; if warranted by the data structure, consider an extended Cox model with a time-varying covariate.

For Experiment 1: if each dog performed only one test, please acknowledge the limitation (no way to estimate a trial trend) and check batch/order at the session level; if multiple tasks/trials occurred per dog, include trial as above. Please provide sensitivity analyses (with vs. without trial) and partial-effect plots.

See: Barr DJ (2013). Random effects structure for testing interactions in linear mixed-effects models. Frontiers in Psychology 4:328.

**Do you want your identity to be public for this peer review?** For information about this choice, including consent withdrawal, please see our Privacy Policy

Reviewer #1: No

Reviewer #2: No

Reviewer #3: **Yes: ** Ivan Khvatov

---

## [Author Response · Author response to Decision Letter 2]

13 Nov 2025

Reviewer #3: Thank you for addressing most of my comments and for the corresponding revisions. A few points, however, remain insufficiently addressed. Below I clarify the rationale and offer specific suggestions for revision.

Point 5 — Demographic covariates

Thank you for the clarification. Here is why we ask you to account for sex and age in the models, even if these variables were not part of the a priori hypothesis.

• “We had no hypotheses about sex/age.” Adding covariates is not about testing a new hypothesis; it is a standard way to control potential confounding and to improve the precision of the main treatment estimate. Modern modeling guidance emphasizes that adjustment for relevant predictors reduces noise and yields more efficient standard errors (SEs) and confidence intervals (CIs). For age, please model the effect flexibly (e.g., restricted cubic splines) rather than as a strictly linear term.

See: Frank E. Harrell Jr., Regression Modeling Strategies.

• “Groups are similar.” Similar means at the group level do not preclude individual-level confounding, particularly when age variance is high; adjusting for informative covariates increases efficiency of the treatment effect even under randomization/balanced groups.

• “All dogs ≥14 months; body image is consolidated.” Physical maturity does not eliminate between-individual differences in attention, learning speed, and cognitive flexibility in adult dogs; age-related variation in these domains is well documented.

See: Wallis LJ, Range F, Müller CA, Serisier S, Huber L, Virányi Z (2014). Lifespan development of attentiveness in domestic dogs: drawing parallels with humans. Frontiers in Psychology, 5:71. https://doi.org/10.3389/fpsyg.2014.00071

• Empirical evidence for sex effects in canine cognition. Several studies report sex differences in dogs’ cognitive/spatial tasks.

See:

• Fugazza C, Mongillo P, Marinelli L. Sex differences in dogs’ social learning of spatial information. Animal Cognition (2017) 20(4):789–794.

• Mongillo P, Scandurra A, D’Aniello B, Marinelli L. Effect of sex and gonadectomy on dogs’ spatial performance. Applied Animal Behaviour Science (2017) 191:84–89.

• Junttila S, Huohvanainen S, Tiira K. Effect of sex and reproductive status on inhibitory control and social cognition in the domestic dog (Canis familiaris). Animals (2021) 11:2448.

• Reporting standards. ARRIVE 2.0 Explanation & Elaboration (Item 7, Statistical methods) explicitly states that nuisance variables should be considered in the analysis (e.g., as covariates); sex and age are listed among factors influencing outcomes. In addition, the NIH SABV policy expects sex to be factored into design, analysis, and reporting in vertebrate animal studies. Please either provide analyses with these covariates or offer a strong justification for excluding them.

See: Percie du Sert N. et al. (2020). Reporting animal research: Explanation and elaboration for the ARRIVE guidelines 2.0. PLOS Biology 18(7): e3000411.

Requested action:

— Refit all generalized linear mixed-effects models (GLMMs) and mixed-effects Cox models (CoxME) including sex and age as fixed covariates; model age with restricted cubic splines (3–5 knots). Report whether the agility-training effect persists and how SEs/CIs change.

— Provide a sensitivity analysis (models with vs. without these covariates); if feasible, exploratory training×age and training×sex interactions with appropriate caveats.

— Reflect these choices transparently in Statistical methods in line with ARRIVE 2.0 (E&E, Item 7).

Point 7 — Fatigue/motivation and trial-order effects

Thank you for the clarification. A subject random intercept adjusts baseline between-dog differences but does not model within-session trends (learning, fatigue, motivational decline). Such trends should be accounted for by an explicit trial index predictor; for inference, please treat trial as a continuous variable with a flexible functional form (e.g., restricted cubic splines) and, where possible, include a by-dog random slope for trial. Person-mean centering of the trial index (within-subject centering) is recommended to cleanly separate within- and between-dog components. This approach is standard for repeated-measures designs and avoids anti-conservative inference; splitting into thirds is unnecessary for the main model and can be reserved for descriptive checks. For time-to-success outcomes, include trial in Cox/CoxME; if warranted by the data structure, consider an extended Cox model with a time-varying covariate.

For Experiment 1: if each dog performed only one test, please acknowledge the limitation (no way to estimate a trial trend) and check batch/order at the session level; if multiple tasks/trials occurred per dog, include trial as above. Please provide sensitivity analyses (with vs. without trial) and partial-effect plots.

See: Barr DJ (2013). Random effects structure for testing interactions in linear mixed-effects models. Frontiers in Psychology 4:328.

Our response:

We agree that age and sex could be added as control variables. We also agree that the effect of age should not be simply modelled as linear. However, we were not sure if the restricted cubic splines (RCS) were the best method to model this effect. We would assume that younger dogs may be worse at assessing the physical parameters of their body due to lack of experience, while this ability may fade for older dogs. Therefore, the dogs that fall in between should outperform the rest. As for the effect of trial, we would actually expect it to be linear, as it is common for dogs to do 20 trials in an experiment, so we would expect some decline in motivation with trial but nothing major. With this in mind, introducing the age squared into the model may be a more accurate way to model the effect of age. Restricted cubic splines often have an issue of loss of interpretability and overfitting the data. To make sure we are using the best possible model for our data, we compared the Akaike Information Criterion (AIC) [1,2]of models using squared term, RCS (4 knots) for age only, and RCS (4 knots) for both age and trial, see table below. The colour fill of the cells represents how well the model fits our data: green representing the best fit (lowest AIC), while the red represents the worst fit (highest AIC), and yellow and orange falling in between.

Colour code

Lowest AIC Highest AIC

Comb12 Original Age squared; sex Age splines (4); sex Age and trial splines (4); sex

App 140.2011 143.545 146.814 151.5677

Att 135.9755 137.5687 140.6845 146.1881

Chosen 138.8667 136.2802 138.164 142.5788

Comb34 Original Age squared; sex Age splines (4); sex Age and trial splines (4); sex

App 119.3054 122.2449 116.2574 119.6225

Att 99.10256 104.4944 103.5049 108.2469

Chosen 78.19002 83.19431 85.3913 89.85315

Comb567 Original Age squared; sex Age splines (4); sex Age and trial splines (4); sex

App 200.5669 204.4066 201.7396 208.3161

Att 131.7965 129.662 126.305 129.4156

Chosen 70.43967 117.9099 119.0174 123.829

Comb8910 Original Age squared; sex Age splines (4); sex Age and trial splines (4); sex

App 191.1069 194.8953 198.3063 201.556

Att 166.6473 172.1097 175.1665 179.193

Chosen 72.37238 137.4145 140.403 145.9742

Solidity Original Age squared; sex Age splines (4); sex Age and trial splines (4); sex

App 1269.479 1271.478 1275.186 1279.921

Att 991.5868 994.0419 997.616 990.9578

Chosen 351.5033 354.0858 356.6618 356.3391

Judging from the AIC comparison, both the model with the square term for age and with RCS for age would be good candidates. However, the square term models seem to have a better fit and paired with the negative sides of RCS, we decided to use the square term models. The models containing z-trial as a RCS yielded much higher AICs, so we decided to keep the z-trial as a linear variable as it both fits our expectations and seems to better fit our data. Additionally, it seems that RCS often overfitted our data. Here is an example of the comparison of effect of age in four different models (original - top left, squared - top right, RCS – bottom left, RCS for age and trial – bottom right).

It is clear that RCS overfitted the data simply because of the requirement to create the number of knots defined. This is usually useful when there are multiple age groups (e.g. in human studies), but for dogs we would only differentiate between young, middle-aged, and old dogs, so introducing more knots would only reduce the interpretability of the results.

We refitted all GLMM and Coxme models with added sex, z-transformed age, and z-transformed age squared. We updated the statistical analysis, results, and discussion(s) to fit the new models. We also explained in the statistical analysis the limitation of our design (each trial being different combination) and why we did not include z-trial for Experiment 1.

References:

1. Zuur AF, Ieno EN, Walker N, Saveliev AA, Smith GM. Mixed effects models and extensions in ecology with R [Internet]. New York, NY: Springer; 2009 [cited 2025 Oct 27]. Available from: https://link.springer.com/10.1007/978-0-387-87458-6

2. Harrison XA, Donaldson L, Correa-Cano ME, Evans J, Fisher DN, Goodwin CED, et al. A brief introduction to mixed effects modelling and multi-model inference in ecology. PeerJ 2018;6:e4794.

---

## [Decision Letter · Decision Letter 2]

25 Nov 2025

Influence of agility training on body-size and object solidity perception in pet dogs

PONE-D-25-25871R2

Dear Dr. Staric,

We’re pleased to inform you that your manuscript has been judged scientifically suitable for publication and will be formally accepted for publication once it meets all outstanding technical requirements.

Kind regards,

Brittany N. Florkiewicz, Ph.D.

Academic Editor

PLOS ONE

Additional Editor Comments (optional):

Reviewers' comments:

Reviewer's Responses to Questions

**Comments to the Author**

Reviewer #3: All comments have been addressed

2. Is the manuscript technically sound, and do the data support the conclusions?

Reviewer #3: Yes

3. Has the statistical analysis been performed appropriately and rigorously?

Reviewer #3: Yes

4. Have the authors made all data underlying the findings in their manuscript fully available?

Reviewer #3: Yes

5. Is the manuscript presented in an intelligible fashion and written in standard English?

Reviewer #3: Yes

Reviewer #3: Dear Authors,

Thank you for your thorough revisions and for addressing the majority of the previous comments. The manuscript has improved substantially and is now suitable for publication in PLOS ONE.

Below are several non-critical editorial suggestions that may further strengthen the clarity and transparency of the work:

Experiment 1 — limitation.

Please briefly acknowledge in the Limitations section that trial-order effects cannot be estimated for Experiment 1, as each dog performed only one test.

Trial-by-subject random slope.

For Experiment 2, including a by-dog random slope for the trial index could be considered. This is a recommendation only and not a requirement.

Optional sensitivity analysis.

If feasible, a short sensitivity comparison of models with vs. without demographic covariates (sex, age) would further demonstrate robustness, although the current analysis is acceptable.

Statistical methods clarity.

It may be helpful to provide a slightly clearer description of the final model structures (e.g., using an lme4-style formula notation).

These suggestions are editorial in nature and do not affect the validity of the study. I recommend acceptance after minor editorial adjustments.

**Do you want your identity to be public for this peer review?** For information about this choice, including consent withdrawal, please see our Privacy Policy

Reviewer #3: **Yes: ** Khvatov Ivan A.

---

## [Editor Report · Acceptance letter]

PONE-D-25-25871R2

PLOS One

Dear Dr. Staric,

I'm pleased to inform you that your manuscript has been deemed suitable for publication in PLOS One. Congratulations! Your manuscript is now being handed over to our production team.

Kind regards,

on behalf of

Dr. Brittany N. Florkiewicz

Academic Editor

PLOS One